METHODS

# A coupling model of transcranial magnetic stimulation induced electric fields to neural state variables

Aaron Miller[1,2], Thomas R. Knösche[1,3], Konstantin Weise[1,4]*

**1** Max Planck Institute for Human Cognitive and Brain Sciences, Leipzig, Germany, **2** Leipzig University, Faculty of Physics and Earth System Sciences, Leipzig, Germany, **3** Technische Universität Ilmenau, Institute of Biomedical Engineering and Informatics, Ilmenau, Germany, **4** Leipzig University of Applied Sciences HTWK, Leipzig, Germany

* kweise@cbs.mpg.de

## Abstract

Transcranial magnetic stimulation (TMS) is a non-invasive brain stimulation technique used to modulate neural activity, with applications in clinical treatment, diagnostics, and neuroscientific research. TMS targeting the human primary motor cortex (M1) is well studied, aided by experimental readouts from the cerebral cortex, the spinal cord, and activated muscle targets. One key readout is a series of pulses that descend the spinal cord following TMS, called DI-waves. These reflect the output of M1 to the spinal cord and are influenced by TMS parameters such as orientation, strength, and waveform. Previous modeling studies have deployed numerous strategies to explain DI-wave generation, but generally approximate TMS inputs as semi-arbitrary current or synaptic inputs. A consistent missing piece to these models is a biophysically motivated coupling between TMS and the neural states of cells and cell populations. This study aims to leverage cable simulations of realistic neuron morphologies to couple TMS induced electric fields to average state variables of cortical cell populations. This coupling model quantifies the spatial-temporal activation function of directly stimulated axonal fibers and the average input current that downstream cells receive due to synaptic inputs from directly stimulated cells. An example M1 cortical circuit is studied, in which TMS stimulates layer 2/3 excitatory and inhibitory neurons that project synapses onto layer 5 corticospinal neurons. Results indicate that TMS induces unique directionally sensitive distributions of synaptic outputs in time and space for each cell type. Directional and dosage sensitivity carries forward to the dendritic current flowing into layer 5 cells. Ultimately, the coupling model provides a novel architecture to translate electric fields from TMS into activation functions that alter neural states and serve as inputs to cortical circuit modeling. The study of other brain regions is achievable through an alternate choice of cell morphologies, cell locations, and circuit design.

**Data availability statement:** All relevant data generated from simulations is available in OSF repository DOI: 10.17605/OSF.IO/NXVDH, https://osf.io/nxvdh/. Code with model core and scripts to generate figures is available at https://github.com/aaron-h-miller/TMS-Coupling-Model.

**Funding:** This work was supported by the BMFTR (01GQ2201 to KW and TRK). The funders had no role in study design, data collection and analysis, decision to publish, or preparation of the manuscript.

**Competing interests:** The authors have declared that no competing interests exist.

## Author summary

Transcranial magnetic stimulation is a powerful technique to interfere with the brain's function in a non-invasive way. It does so by magnetically inducing an electric field in the brain that influences the state of neurons. It has widespread medical application for diagnostics and treatment of diseases, such as depression. Moreover, it is valuable for fundamental brain research as it allows stimulating the brain and then observing its response. In order to optimize its utility in medicine and drawing more well-founded conclusions from scientific experiments, it is important to precisely describe its working mechanisms. In this work, we introduce a computational technique to simulate a crucial ingredient of these mechanisms, namely the precise way the induced electric field excites populations of interconnected neurons. We exemplify the technique for the case of stimulating the motor areas of the cortex, causing muscle activity. However, it can be generalized to a wide variety of use cases, thus forming an important tool that helps researchers to devise accurate mechanistic links between transcranial magnetic stimulation and its observed effects. In consequence, it supports future research aiming at the development of better diagnostics and treatment of brain diseases, as well as better understanding of brain function.

## Introduction

Non-invasive brain stimulation is a powerful set of tools that can modulate brain activity through the safe application of electromagnetic fields and without the need for surgery or implants. TMS offers targeted modulation of neural activity giving rise to immediate and long-lasting effects, making it valuable for probing and influencing the nervous system for diagnostic, therapeutic, and scientific purposes. However, in order to draw accurate conclusions about TMS activation and behavioral effects, we must first understand the mechanistic relationship between stimulation and observable readouts. While modeling approaches may entail variable chains of computation steps, depending on the stimulated brain area and the considered readout, they all rely on one crucial ingredient, namely the precise way the induced electric field excites populations of interconnected neurons. In this study, we develop a generic computational technique to tackle this step and study it using the example of TMS of the primary motor cortex (M1).

M1-TMS is widely studied through behavioral experiments and measurements of electrophysiological readouts. Experimental measurements of the brain, spinal cord, and activated muscles enable the study of structure-function relationships in the motor system as well as pre-surgical planning and clinical diagnostics. The mechanism between TMS stimulation of M1 and muscle contraction is a complex cascade of interactions summarized in Fig 1. Foremost, TMS operates by the delivery of a sharply time-varying current through an induction coil that in turn generates a time-varying magnetic field inside the head. The magnetic flux flowing through

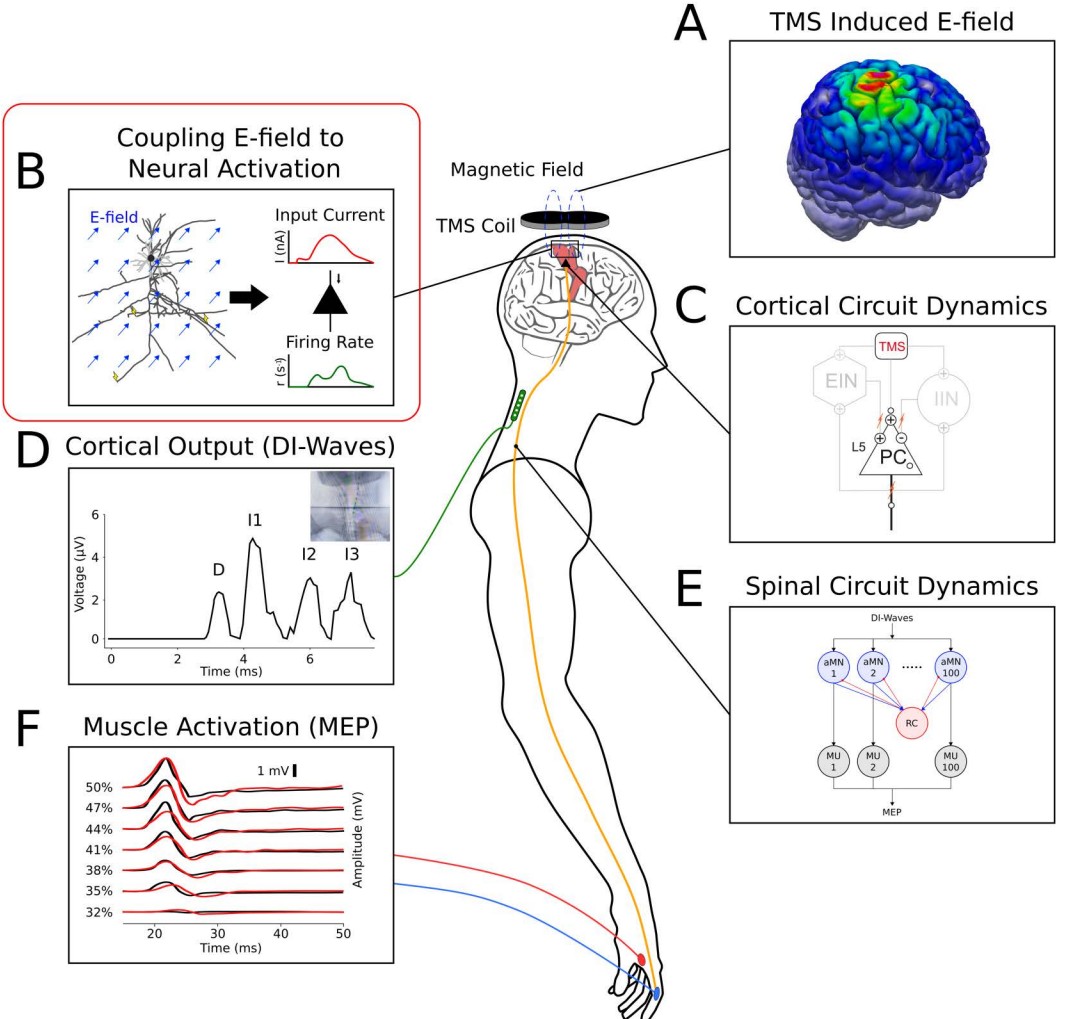

**Fig 1. Full Motor TMS Cascade. (A)** TMS induced electric fields in the cerebral cortex. **(B)** Coupling between induced electric fields and the states of cortical neurons (the focus of this work). **(C)** Circuit dynamics of cell populations in M1 following stimulation. The cortical output is determined via the firing behavior of L5 pyramidal cells projecting to the spinal cord. **(D)** Implanted epidural electrode recording of DI-waves descending the spinal cord following TMS, which are modulated by stimulation conditions. **(E)** Spinal cord microcircuit of alpha motor neurons ($\alpha$MNs) with global inhibition from Renshaw cells (RCs). $\alpha$MNs send action potentials through peripheral nerves to activate motor units at the muscles. **(F)** Traces of motor evoked potentials (MEPs) at increasing stimulation intensities, measured by skin mounted electrodes. Motor units of the muscles are excited by action potentials from spinal motor neurons, causing contraction and a quantifiable MEP readout.

cortical tissue induces a secondary electric field strong enough to depolarize neurons. The TMS coil position, coil geometry, current waveform, head geometry, and tissue conductivity determine the spatial and temporal patterns of induced electric fields, which can be modeled using numerical field (e.g., finite element) calculations of Maxwell's equations [1] (Fig 1A). Second, induced electric fields influence the states of neurons by increasing excitability and directly generating action potentials on axon segments (Fig 1B). Third, this alteration in neuron behavior and state modulates the dynamics of networks of interconnected populations of cells (Fig 1C). Fourth, as action potentials from M1 travel down to spinal motor neurons, volleys of action potential arrivals can be recorded with implanted epidural electrodes (Fig 1D) [2–4]. These pulses are referred to as a direct wave (D-wave) and indirect waves (I-waves) for their hypothesized origins [5].

Corticospinal tract pyramidal cells (PCs) embedded in deep layer 5 (L5) of M1 carry signals from M1 to the spinal cord and the periphery, thereby acting as the output node of the cortex. D-waves are believed to be the result of direct stimulation of fiber tracts extending from L5 to the spinal cord, while I-waves are thought to stem from indirect activation of L5 cells via synaptic connections from more superficial neurons in the M1 cortical microcircuit [3]. I-wave-like frequency deflection is also measurable by EEG, reflecting bursts of activity within M1 [6]. Fifth, subcircuits in the lower cervical spinal cord, composed of alpha motor neurons and inhibitory Renshaw cells, respond to the incoming action potentials and send new action potentials to peripheral nerve fibers projecting to the muscles (Fig 1E). Sixth, motor units are excited by the incoming action potentials, activating and contracting muscle fibers. Skin mounted electromyographic (EMG) electrodes record motor evoked potentials (MEPs) immediately following TMS (Fig 1F). Precise techniques for mapping MEPs have also been developed, localizing the cortical representations of muscle targets in M1 [7,8].

The focus of this study is in the second and third stages of the M1 TMS cascade described above: the direct activation of cells within the cortex and modulation of neuron states in interconnected networks. Since I-wave generation is attributed to intracortical dynamics that project to L5 PCs, this work centers on developing novel modeling strategies for coupling TMS induced electric fields to cortical neural states that govern I-wave generation. Previous studies, in contrast, generally approximate TMS inputs to cortical populations by arbitrary current pulses or fractional cell activation that do not consider directionally sensitive electric field interactions with neurons (see Background). Therefore, this work, referred hereon as the *coupling model*, provides a missing link in the mechanism translating TMS induced electric fields to neural states. The firing pattern of the cortical output neurons (L5 PCs) is determined by the time evolution of their somatic membrane potentials, which are themselves driven by synaptic inputs caused by upstream activation from TMS. The input that postsynaptic somata receive is shaped by the activation pattern of upstream axons by TMS, the geometry of upstream axonal arbors, the distribution of synapses on postsynaptic dendrites, and the geometry of postsynaptic dendrites. As described in detail in Methods, the coupling model simulates reconstructed neuron morphologies to quantify activation functions of upstream cells directly stimulated by TMS. Synaptic and dendritic integration of L2/3 inputs then determines the driving current to L5 PC somata, which ultimately modulates their neural states.

The methods developed here seek to quantify the activation function between synaptically coupled populations under which the presynaptic cells are directly activated by TMS. In the construction of a cortical microcircuit, the coupling model is thereby applied between every set of coupled cell types given the cell type spatial distributions, their morphologies and biophysics, and an estimate of the electric field they experience. The core impact of this model facilitates realistically motivated coupling between TMS induced electric fields and mean field state variables. Choosing an alternate set of cell morphologies, electric fields, and cortical circuits will generalize the coupling model to other forms of electromagnetic brain stimulation in other regions than M1. Fundamentally, this model seeks to determine the under-explored role of the electric field interaction with cortical neurons in the TMS modulation of neural states. This work foremost develops the coupling methods, as well as explores an example deploying those methods to compute the activation functions of different cell classes and the synaptic projection between L2/3 and L5 of M1.

## Background: The state of the art in coupling TMS to cortical models

The last decades have seen a diverse set of models developed to explain and reproduce readouts like DI-waves and MEPs induced by TMS of M1. Three major classes of simulation methods include spiking neural networks, compartment modeling, and neural mass modeling. For example, Esser (2005) considers a spiking neural network of tens of thousands of integrate-and-fire neurons with millions of connections, forming the primary motor cortex and thalamus [9]. M1 receives inputs from the thalamus, premotor, and somatosensory areas. This model introduces TMS as a homogeneous activation of a fraction of all fiber terminals across M1 (between 0 and 100%), assuming fiber orientation is not a factor in the direct response of cell populations. In practice, this strategy results in a fraction of point neurons receiving simultaneous synaptic inputs due to TMS. Yu (2024) extends this spiking network approach by reducing model complexity and removing the

thalamus, which has been shown not to contribute to I-wave generation [10,11]. Stimulation is thus represented as a fraction of cells and long-range fibers that simultaneously fire a single action potential to all synapses they are connected to.

Rusu et al. (2014) (extended by [12,13]) developed compartment modeling based approaches [14]. They consider a population of 300 L2/3 single compartment neurons (excitatory and inhibitory) that project onto one morphologically realistic L5 PC. These models explore dendritic integration and the role of differential positions of synaptic inputs along the dendritic tree as the source of the I-wave volleys arriving in short intervals at the soma. TMS enters the system as a direct current injection to a fraction of the 300 L2/3 excitatory and inhibitory cells (alongside direct activation of the L5 PC). Each activated L2/3 neuron elicits a single spike projecting to positions distributed on the L5 PC dendrites.

A third example by Wilson (2021) leverages neural mass modeling [15]. M1 is abstracted to characteristic populations of L2/3 excitatory interneurons and L2/3 inhibitory interneurons feeding forward to a L5 pyramidal cell population. The state of each neural mass is represented by the average firing rate of all cells in each population. TMS enters the system as a square pulse in input rate (action potentials per second) for 0.5 ms introduced along each axon in the circuit. L2/3 excitatory interneurons were activated with the input rate pulse whose height follows a sigmoid function with respect to a normalized TMS intensity parameter. L5 PCs were activated at 10% lower amplitudes than L2/3 excitatory cells. L2/3 inhibitory neurons were provided a fixed amplitude input.

All of these modeling styles explore differing mechanisms of DI-wave generation (and/or MEPs) with notably similar assumptions about stimulation input. Namely, neurons are preferentially activated by electric fields on axons at geometric discontinuities like bends and terminals. This is supported by experimental studies on nerve fibers [16] and modeling studies of nerves and reconstructed neurons [17–20]. The models described above therefore all consider TMS input at the axons of each stimulated cell, or correspondingly at the synapses of downstream cells. They assume, however, that only a single action potential is generated simultaneously at every stimulated cell. Additionally, all models ignore or simply approximate the directional sensitivity of neuron activation. Fibers are most sensitive to activation when the electric field is parallel to the fiber axis, as cable theory indicates that extracellular stimulation produces depolarization proportional to the parallel component of the electric field gradient [19,21–23]. Thereby, these models are unable to consider how patterns in axon terminal orientation play a role in defining directional sensitivity of TMS activation.

The coupling model developed in this work bases upon the work of Weise and colleagues (2023), which showed that TMS induced action potentials are generated preferentially at axon terminals with directional sensitivity unique to different cell classes and stimulation intensities [23]. A key extension in this study, however, is that upon stimulation, action potentials propagate back through the axons to eventually reach other axon terminals. Regardless of the exact position or number of positions where action potentials are generated by TMS, there is a spatial-temporal process of backpropagation that differentiates the times and locations of action potential arrival at downstream synapses. This means that each stimulated cell outputs many action potentials in a distribution of times and locations, not just a single action potential instantaneously after TMS. The goal of this coupling model is not to directly produce I-waves or MEPs, but to explore how activation by TMS induced electric fields alters ongoing firing rates (state variables) of different neuron populations across different stimulation intensities and orientations. Ultimately, this model then can be deployed in similar studies described above to define the TMS activation of cortical circuits.

## Methods

### Modeling pipeline

Fig 2 depicts an overview of the TMS coupling model as generalized to be computed between a presynaptic and a postsynaptic population in a synaptic connection. This work considers the coupling from L2/3 excitatory and inhibitory populations to L5 as a case study. As discussed in Background, compartment simulations show that TMS induced electric fields generate action potentials on the axons of cortical cells and introduce directional and dosage sensitivity [20,23]. TMS induced electric fields therefore enter the coupling model as an input, interacting directly with the axons of the presynaptic

PLOS Computational
Biology

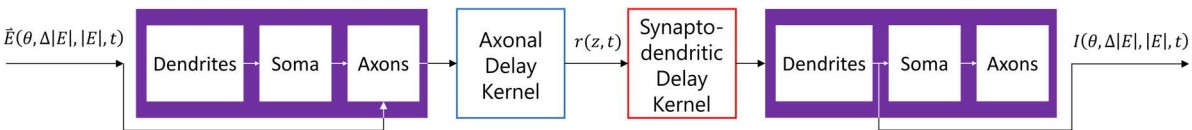

**Fig 2. Modeling flowchart of TMS coupling.** Electric fields enter as the input to a presynaptic populations, coupling directly to axons. The axonal delay kernel simulation quantifies the spatiotemporal output of the presynaptic populations as a density function $r(z, t)$. This density function then enters the (synapto-) dendritic kernel simulation as an input to drive dynamics on the postsynaptic dendrites. Dendritic current entering the soma of the postsynaptic cells exits the model as the output measure, inheriting sensitivity to direction and dosing implicitly from the electric field input.

cells. In the Axonal Delay Kernel Section, the application of electric fields is described, which produces the average spatial-temporal distribution of output spikes from the presynaptic population, forming a kernel $k(z, t)$, where $z$ denotes the position along the cortical depth axis and $t$ is the time from the TMS pulse. These simulations are referred to as the *axonal delay simulation* or *axonal delay kernel*. The axonal delay kernel directly defines the distribution of synaptic inputs that postsynaptic cells receive to their dendrites. The simulation on postsynaptic cells is referred to in this work as a *synapto-dendritic delay simulation*, *synapto-dendritic delay kernel*, or simply *dendritic delay kernel*. The model output is the averaged dendritic current entering the postsynaptic soma. The coupling model is formulated between presynaptic and postsynaptic populations due to the fact that neural activation occurs on cell axons, thereby in the middle of the synaptic pathway between two populations. This allows directional sensitivity, dosage, and TMS waveform, i.e., the electric field induced by TMS, to directly drive neural state variables through estimated input currents.

## Neuron models

This study was greatly aided by the simulation protocol provided by Weise (2023) [23], in which reconstructed rat neuron morphologies from the Blue Brain Project [24] are loaded into the NEURON software package [25]. The coupling model includes 24 layer 2/3 (L2/3) thick tufted pyramidal cells (PCs), 30 layer 5 (L5) pyramidal corticospinal tract neurons, 70 layer 4 (L4) small basket cells (SBCs), 70 L4 nested basket cells (NBCs), and 105 L4 large basket cells (LBCs). All cells are adjusted for shrinkage, smoothed, branch connectivity is repaired, and cell scale and biophysical parameters are modified to be consistent with human neurons (see [23] and [19]). All cable simulations are performed at $37°C$ with initial membrane potential -70 mV. To ensure all simulations begin at steady-state, all simulations are first run for $10^{11}$ ms with a $10^9$ ms step size to allow the system to settle before any stimulus is applied. In the NEURON simulation environment (version 3.9.19) interfaced via Python (version 3.9.19), morphologies were modeled as quasi one-dimensional cables with the inclusion of both active and passive biophysical properties including extended Hodgkin-Huxley-like ion channel models based on human neurophysiology. The morphologies are broken into straight sections of cell tissue coupled together, with each section broken into smaller compartments called segments. Segments serve as the discrete computational units whose membrane potential are simulated and coupled to neighboring segments by capacitive, resistive, and synaptic trans-membrane and axial currents. Electromagnetic fields induced by brain stimulation generate a local extracellular pseudo-potential [26], which couples to the cable equations of individual neuron compartment models by integrating the electric field component along the axial direction of each compartment [23]. Hodgkin-Huxley dynamics and cable theory drive action potential generation under electric field application and the propagation of action potentials on the axonal arbors and dendritic trees of these compartment models. For more detail on the modeled biophysical parameters, neuron morphology generation, and synapses see [19].

## Axonal delay Kernel

**Coupling electric fields to NEURON simulations.** The axonal delay simulation is an extension of existing models that couple electric fields to neural activation [19,20,23]. Each neuron is exposed to electric fields of varying directions

and strengths, parameterized in spherical coordinates. As depicted in Fig 3A., the polar angle $\theta$ quantifies the angle between the electric field and the somatodendritic axis (z'-axis) ranging from 0° to 180°. The azimuthal angle $\varphi$ quantifies the electric field direction in the plane perpendicular to the somatodendritic axis ranging from 0° to 360°. The orientation of cells in the cortex is random about the azimuthal axis, therefore simulations are always averaged across 60 discrete values of $\varphi$ between 0° and 354° in 6° steps. The electric field intensity ($|\mathbf{E}|$) quantifies the factor that scales the normalized TMS coil current waveform (Only biphasic TMS was explored in the example studied here). The electric field gradient ($\triangle|\mathbf{E}|$) quantifies the relative change in field magnitude per unit length parallel to the somatodendritic axis (Also due to scope and computational time, $\triangle|\mathbf{E}|$ was fixed to 0% in this work).

Each axonal delay simulation applies an electric field and results in zero, one, or more action potentials to be generated at or near axon terminals. Action potentials then backpropagate along the axonal arborizations until reaching remaining axon terminals, each of which projects to synaptic targets elsewhere in the cortical microcircuit (S1 Video, Fig 3B). The exact spatial and temporal dynamics of these action potential arrivals are unique to every cell morphology and stimulation paradigm, controlled by non-linear interactions of one or more action potentials on the axonal arbor. In this way, the cell morphology and applied field parameters dictate the spatial and time dynamics of its synaptic outputs. For each terminal which received an action potential, the arrival time with respect to stimulation onset as well as the depth of the terminal (z'-axis coordinate) with respect to the soma is recorded in a histogram (Fig 4). Due to translational and rotational symmetry of neurons in the cortex as well as its laminar organization, only the vertical axis coordinate is significant to the average coupling between cell populations within the column. Performing this simulation for a wide range of field parameters maps directional and intensity sensitivity of the electric field to synaptic outputs in space and time. The profile defined by the two-dimensional (2D) histogram of axon terminal arrival times per value of (z', t) is referred to as an *axonal delay kernel*.

**Mean field calculation of cortical population excitation.** Every simulation is averaged across 60 unique rotations of each morphology about its somatodendritic axis, yielding an average axonal delay kernel for a given cell morphology and set of electric field parameters $(\theta, \triangle|\mathbf{E}|, |\mathbf{E}|)$. Axonal delay simulations are also averaged across all morphologies within a

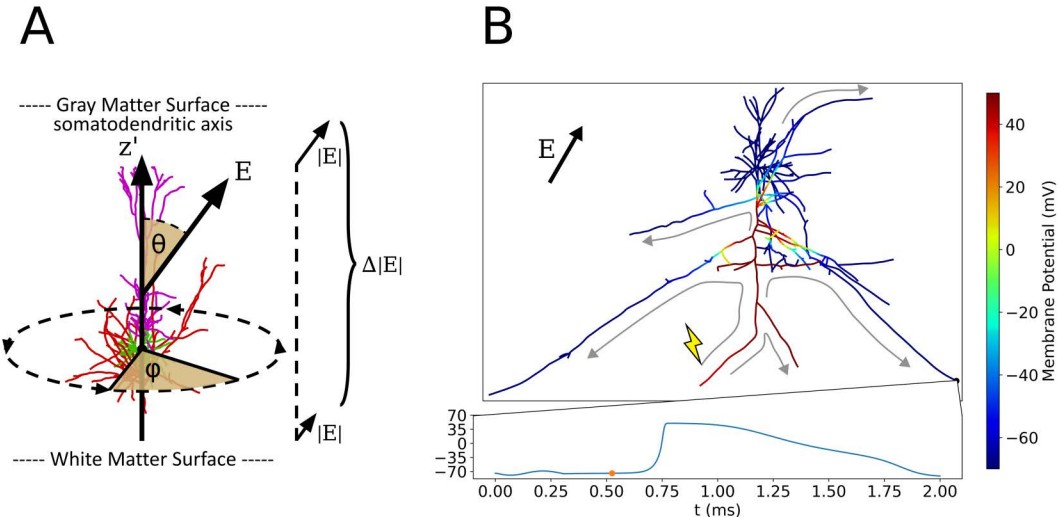

**Fig 3. Action potential backpropagation after application of parameterized electric field. (A)** Electric field parameters in relation to cell morphology: Polar angle $\theta$, electric field intensity gradient $\triangle|\mathbf{E}|$, azimuthal angle $\varphi$, and electric field intensity $|\mathbf{E}|$ that scales the normalized waveform (Mono-/Biphasic, etc.). **(B)** Single frame at t = 0.525 ms of axonal delay simulation from S1 Video. Electric field parameters are $\theta$ = 30°, $\triangle|\mathbf{E}|$ = 0%, $\varphi$ = 0°, $|\mathbf{E}|$ = 225 V/m, biphasic pulse. The color axis is the membrane potential of each compartment. One axon terminal is activated initially in this simulation at the lightning bolt and the depolarization backpropagates to the remaining axonal compartments as depicted by the gray arrows. The membrane potential time series below is shown for the axon terminal at the bottom right, which receives an action potential around 0.75 ms.

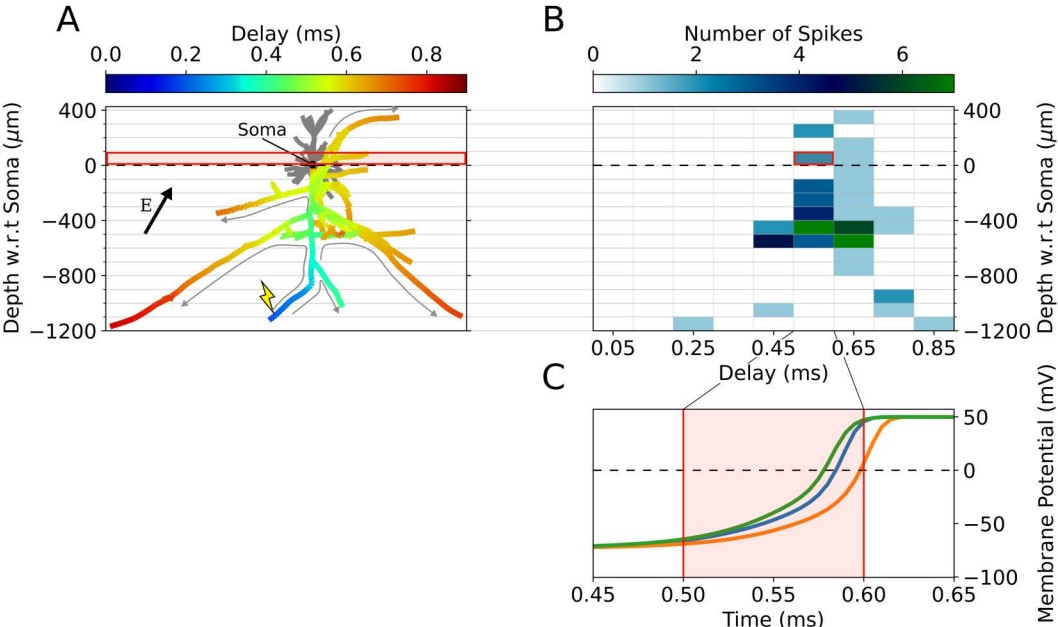

**Fig 4. Example Axonal delay histogram calculation. (A)** Action potential backpropagation for electric field parameters $\theta = 30°$, $\Delta|\mathbf{E}| = 0\%$, $\varphi = 0°$, $|\mathbf{E}| = 225 V/m$, and biphasic pulse (Cell activation threshold is 140 V/m). Color axis indicates the arrival time of an action potential at the associated axon compartment in ms. Activation in this example occurs only at the terminal marked by the lightning bolt, with gray arrows depicting the backpropagation through the axonal arbor to other axon terminals. Boxed in red is an example z-bin between $z = 0$ $\mu$m and $z = 100$ $\mu$m. **(B)** Two-dimensional histogram of action potential arrival times and locations for each axon terminal in this cell. Boxed in red is a chosen histogram bin (with same z-values as in **(A)**) with depth between 0 and 100 $\mu$m and time between 0.5 and 0.6 ms. **(C)** Membrane potential timeseries for the three axon terminals whose action potentials arrive within the time-depth-bin boxed in **(B)** (i.e., their position lies within the depth bin and their potentials cross $V = 0$ $mV$ within the time bin). The boundaries of the bin in the time axis are also boxed in red.

given cell type in order to reflect mean dynamics representative of trends in cell type-specific geometry. The axonal delay kernel is computed for excitatory L2/3 PCs, inhibitory L2/3 BCs, and excitatory L5 PCs. The response of inhibitory L2/3 basket cells (BCs) is computed from basket cell morphologies originally recovered from L4 of the somatosensory cortex, artificially placed at the same distribution of soma locations as L2/3 excitatory cells. Since basket cells are modeled across small, nested, and large basket cells for different electric types, the average inhibitory cell kernel is an average of independent kernels from each subtype and electrical type, weighted by their measured relative frequencies in the cortex (normalized frequencies reported in Table 1).

After averaging the axonal delay kernel histograms across each azimuthal angle, each cell morphology, and across cell subtypes from Table 1, we end up with a histogram representing the average distribution of spike outputs in time and depth with respect to the soma. Fig 5 depicts the averaging processes for the case of L2/3 PCs, while this process is also carried out for L5 PC and L4 BC morphologies. Additionally, cells within a given layer are placed at a range of soma depths within the cortex. To account for this variability, the 1D cross correlation is computed between the axonal delay kernel for the average cell and the function describing the distribution of cell soma positions with respect to cortical depth (z-axis). This produces the axonal delay kernel for a population of distributed cells (Fig 5D). The axonal delay kernel of L2/3 excitatory and L2/3 inhibitory populations projecting to L5 are then scaled by the ratio of cell counts of presynaptic/postsynaptic cells. The cell count ratios are computed from multi-method classification for L2/3 PC/L5 PC as 5.04, and L2/3 BC/L5 PC as 0.40 [28]. Inhibitory cells were classified by expression of Gamma-aminobutyric acid (GABA) receptors but not categorized by layer, thus all GABAergic cells lying between normalized cell depths of 0.1 to 0.29 (-270 $\mu$m to -783

**Table 1. Cell types and relative frequency per layer.** Relative frequency weighting factors recovered from cell morphology and electric type resolved survey of rat somatosensory cortex via the Blue Brain Project online microcircuit explorer [24,27].

| ⟨ Layer ⟩–⟨ Morphological Type ⟩–⟨ Electrical Type ⟩ | Relative Frequency in Layer |
|---|---|
| L2/3_PC_cADpyr | 1 |
| L5_TTPC2_cADpyr | 1 |
| L4_LBC_dNAC | 0.245 |
| L4_LBC_cSTUT | 0.165 |
| L4_LBC_cACint | 0.117 |
| L4_NBC_dNAC | 0.191 |
| L4_NBC_cACint | 0.053 |
| L4_SBC_bNAC | 0.112 |
| L4_SBC_cACint | 0.117 |

Pyramidal Cell (PC), Large Basket Cell (LBC), Nested Basket Cell (NBC), Small Basket Cell (SBC). Electrical subtypes are described in [24].

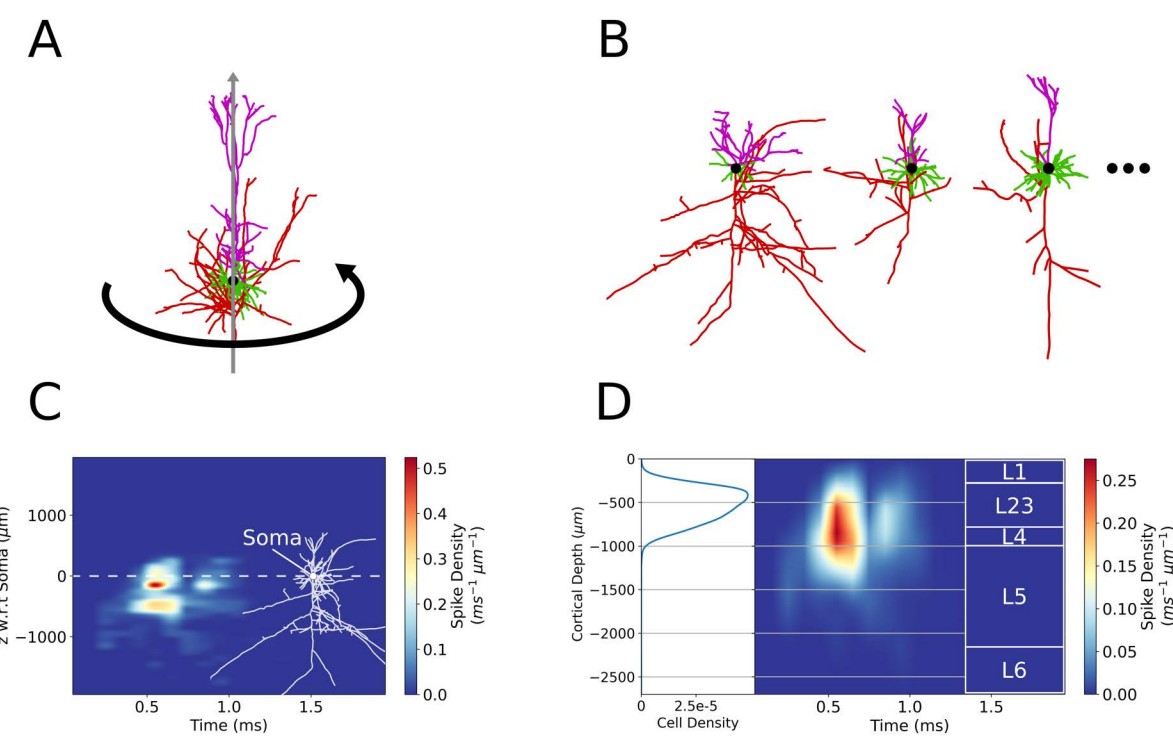

**Fig 5. Averaging of axonal delay simulations. (A)** Visual representation of averaging across azimuthal angle. **(B)** Visual representation of averaging across unique cell morphologies. **(C)** Average L2/3 PC axonal delay kernel for electric field parameters $\theta = 30°$, $\Delta|\mathbf{E}| = 0\%$, and $|\mathbf{E}| = 225$ V/m. Overlaid at right is a L2/3 PC morphology with soma aligned to the vertical axis and a horizontal dotted line at the soma depth ($z' = 0$). **(D)** Average L2/3 population axonal delay kernel calculated by correlation of the average cell kernel **(C)** with the cell density distribution [28] (left) and scaled by L2/3 PC/L5 PC cell frequency ratio. Overlaid at right are the M1 cortical layer boundaries.

$\mu$m) were counted. These layer boundaries for L2/3 were estimated from studies of primary motor cortex architecture with the inclusion of the recently discovered thin layer 4 in M1 [28–31].

The population axonal delay kernel is computed numerically from the cell average as follows. Let the average spike density distribution with respect to the cell soma be represented by $r(z', t)$. The variable $z'$ notates position with respect to the cell soma in the axis parallel to the somatodendritic axis where $z' \in [-2000, 500]$ $\mu$m in steps of 100 $\mu$m and total vector length $N_r$ (Positive direction is oriented from the soma toward the dendrites. Exact ranges vary between morphological types due to cell sizes). $z' = 0$ $\mu$m lies at the cell soma. The cell density function $d(z)$, however, is in the domain of the cortical depth axis (z-axis), where variable $z$ represents position in the cortical column. $z \in [-2700, 0]$ $\mu$m in steps of 100 $\mu$m and total vector length $N_d$. $z = 0$ $\mu$m lies at the CSF boundary with layer 1, while $z = -2700$ $\mu$m lies at the layer 6 to white matter boundary at the bottom of the cortical column. The correlation of spike density distribution with the cell density function is executed only along the z-axis (which is parallel to the z'-axis) for every value of $t$. The time (t-axis) is unaffected by the correlation operation. SciPy's cross-correlation function (version 1.13.1) under the 'same' mode has an output the same length as the cell density function and lies in the same domain as the cortical depth axis z-axis. The correlation process amounts to a change of basis from $z' \rightarrow z$, i.e., from position with respect to cell soma to position with respect to cortical surface.

The cell density function $d(z)$ must also be normalized such that

$$\sum_i d_i = 1$$

and thereby the volume under the axonal delay kernel, thereby the number of spikes, is preserved by the correlation operation. The computation of the population axonal delay kernel $k(z, t)$ is described as:

$$k(z, t) = \text{corr}\left(d(z), \ r(z', t)\right)$$
$$k_{ij} = \sum_n d_{n+i} \cdot r_{nj}$$

$$(1)$$

where the index $i$ corresponds to the z-axis and index $j$ to the t-axis. The full discrete cross-correlation performs the sum with index $n$ up to a length $N_d + N_r - 1$ across the z/z' axes. The 'same' mode however crops the result to $N_d$, centered with respect to the full correlation. The end result thereby has the same length as the cell density function and lies in the domain of the cortical depth (z-axis) with discrete index $i$. In other words, the correlation performs a transformation from the z'-axis to the z-axis. The correlation approach also makes the implicit assumption that cells at slightly different depths ($\pm 0.5$ mm) within a given layer experience the same electric field. Cell density distributions are calculated from cell cortical depth values recorded from an extensive spatially resolved classification study of the primary motor cortex [28]. The cell density distribution is shown in Fig 5D for L2/3 PCs. When calculating the L2/3 inhibitory population kernel, the average cell kernel computed for L2/3 BCs (originally recovered from L4 of rat somatosensory cortex) is correlated with the same cell density function as that from the L2/3 PCs. Because these cell density curves were recovered from rat brain imaging, the cortical depth axis was normalized and re-scaled such that the cortical column (z-axis) has a total depth of 2700 $\mu$m, as measured for the human primary motor cortex ex vivo [32]. The resultant kernel $k(z, t)$ thus quantifies the average output of cells of the same type distributed across cortical depth upon activation by TMS. Kernel histograms are then interpolated with a pchip method as in Fig 5 and in the axonal delay figures in Results. The precomputation of axonal delay kernels for L2/3 PC, L5 PC, and 7 L4 BC subtypes were simulated on 10 cluster nodes with 72 cores each (Intel Xeon IceLake-SP 8360Y, 256 GB RAM) for approximately 7 days.

## Synapto-dendritic delay Kernel

**Coupling pre- and postsynaptic simulations.** As TMS induced action potentials backpropagate through the axonal arbor of a cell, these action potentials inevitably reach axon terminals and synapse to other cells. To model the current input that a postsynaptic soma receives due to lying downstream of TMS activated cells, the inputs to the postsynaptic cell are determined by the axonal delay kernels of the upstream populations.

Foremost, for each set of electric field parameters, the axonal delay kernels of the L2/3 PC and BC populations determine the spatial-temporal distribution and magnitude of synaptic inputs to L5 dendrites. We then model these synaptic inputs as continuous conductance functions for four synapse receptor types directly computed from the axonal delay kernels embedded at each compartment on the L5 dendritic tree. The resulting dynamics is simulated to model the current entering the soma due to passive and active properties of postsynaptic potential evolution as well as dendrite geometry. The synapto-dendritic current is directly measured in-silico as the axial current at the interface between L5 PC dendrites and the soma. This simulation process is repeated and averaged across 30 different L5 PC morphologies as well as 25 L5 cell positions to characterize the average input received by all of L5 from L2/3.

Fig 6 depicts an exemplar individual synapto-dendritic delay simulation, which models synaptic inputs and resulting postsynaptic potential dynamics on a single representative L5 PC placed at the average L5 soma depth. Dendritic delay simulations reported in the Results are run with a time step size of 0.01 ms for 100 ms. Fig 6 is cropped to 50 ms.

For each value of cortical depth ($z$) and time ($t$) (with bin sizes $\Delta z = 50$ $\mu$m, $\Delta t = 0.01$ ms), the axonal delay kernel $k(z, t)$ (Figs 6A and B) determines the density of input spikes arriving from L2/3 BC and L2/3 PC presynaptic populations. Each dendritic compartment within each bin $z \in [z - \Delta z/2, z + \Delta z/2]$ receives four input currents due to $\alpha$-amino-3-hydroxy-5-methyl-4-isoxazolepropionic acid (AMPA), N-methyl-D-aspartate (NMDA), Gamma-aminobutyric acid (GABAa and GABAb) receptor synapses (Fig 6H). Trans-membrane currents are computed from conductance via Ohm's law for conductance based synapses:

$$I_{mem} = g(v - E_{reversal}) \tag{2}$$

where $g$ is the synaptic conductance, $v$ is the membrane potential of the compartment (simulated state variable), and $E_{reversal}$ is the reversal potential of the synapse. Synaptic rise and decay time constants, reversal potentials, and receptor-specific kinetics were adopted from previously published biophysically detailed cortical models [14,33], where they were constrained by experimental electrophysiological measurements and shown to reproduce physiologically realistic EPSC and IPSC dynamics. The parameters are given in Table 2.

Note that NMDA conductance is additionally scaled by the gating factor $1/\left(1 + 0.28 * Mg * exp(-0.062 * v)\right)$ where Mg is the concentration of extracellular magnesium, set here to 1 mmol. The synaptic conductance $g$ must be computed for each synapse type and each dendritic compartment respectively. The AMPA, NMDA, GABAa, and GABAb conductances for a single compartment are computed as:

$$g_{AMPA} = \frac{A}{A_{bin}} (1 - f_{NMDA}) f_{ex} K_{AMPA}(t) \circledast K_{z,ex}(t) \tag{3}$$

$$g_{NMDA} = \frac{A}{A_{bin}} f_{NMDA} f_{ex} K_{NMDA}(t) \circledast K_{z,ex}(t) \tag{4}$$

$$g_{GABAa} = \frac{A}{A_{bin}} f_{GABAa} (1 - f_{ex}) K_{GABAa}(t) \circledast K_{z,inh}(t) \tag{5}$$

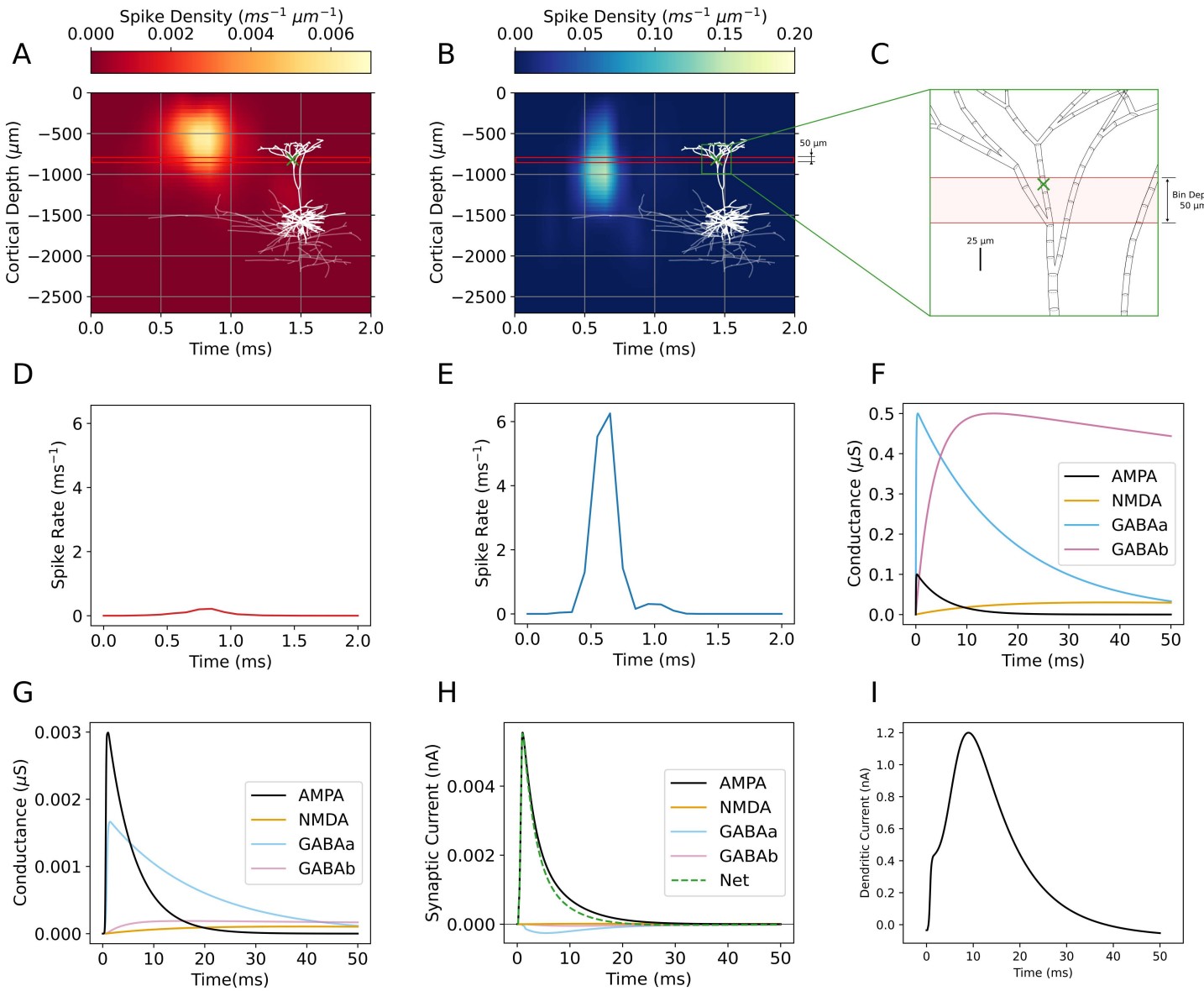

**Fig 6. Synaptic inputs and model output of a single example synapto-dendritic delay simulation.** ($\theta$ = **30°**, $\Delta|\mathbf{E}|$ = **0%**, $\varphi$ = **0°**, $|\mathbf{E}|$ = **225** V/m, 0.01 ms step size simulated for 50 ms). **(A)** Inhibitory axonal delay kernel (L2/3 BC population). **(B)** Excitatory axonal delay kernel (L2/3 PC population). Overlaid over both kernels is the postsynaptic L5 PC with soma at -1584 $\mu$m, an example dendritic compartment marked by a green cross, and the bin containing that compartment boxed in red. **(C)** Zoomed-in view of L5 PC morphology near the green cross. **(D)** Partial integral of the inhibitory axonal delay kernel across the red boxed, 50 $\mu$m bin in **(A)**. **(E)** Partial integral of the excitatory axonal delay kernel across the red boxed, 50 $\mu$m bin in **(B)**. **(F)** Double exponential conductance kernels describing AMPA, NMDA, GABAa, GABAb synapses. **(G)** Synaptic conductance of each synapse type at green cross-marked compartment, computed by the weighted convolution of **(D)** and **(E)** with the kernels in **(F)**. **(H)** Simulated trans-membrane current of each conductance-based synapse at green cross-marked compartment. Positive valued current flows into the cell. **(I)** Dendritic current flowing into the L5 PC soma from the dendrites due to synaptic inputs.

**Table 2. Double Exponential Mechanism Constants. Synaptic time constants, reversal potential, and peak conductance for the 2 excitatory (AMPA/NMDA) and 2 inhibitory (GABAa/GABAb) synapse types used.**

| Type | Receptor | $\tau_{rise}$ (ms) | $\tau_{fall}$ (ms) | $E_{reversal}$ (mV) | $g_{peak}$ ($\mu$S) |
|------|----------|--------|--------|---------|---------|
| Excitatory | AMPA | 0.05 | 5.3 | 0 | 0.1 |
|  | NMDA | 15 | 150 | 0 | 0.03 |
| Inhibitory | GABAa | 0.07 | 18.2 | -80 | 0.5 |
|  | GABAb | 3.5 | 260.9 | -93 | 0.5 |

Reversal potentials and peak conductances from [14], time constants from [33].

$$g_{GABAb} = \frac{A}{A_{bin}} (1 - f_{GABAa})(1 - f_{ex}) K_{GABAb}(t) \circledast K_{z,inh}(t) \tag{6}$$

Where,

$$K_{SYN} = g_{peak} N \left( e^{-t/\tau_{fall}} - e^{-t/\tau_{rise}} \right) \tag{7}$$

$$N = e^{-t_{max}/\tau_{fall}} - e^{-t_{max}/\tau_{rise}}, \text{ with } t_{max} = \frac{\tau_{rise}\,\tau_{fall}}{\tau_{fall} - \tau_{rise}} \ln \left( \frac{\tau_{fall}}{\tau_{rise}} \right)$$

$$K_{z,ex/inh}(t) = \int_{z-\Delta z/2}^{z+\Delta z/2} k_{ex/inh}(\tilde{z}, t) \, d\tilde{z} \tag{8}$$

Each synaptic conductance (Eqs 3–6) is computed by convolving the double exponential kernels (Eq (7) and Fig 6F) with the partial integral of the axonal delay kernels across the bin of width $\Delta z = 50$ $\mu$m (Eq (8) and Fig 6D and 6E). Double exponential kernels are defined by peak conductance and rise and decay times from existing compartment modeling studies [14,33] (see Table 2).

For AMPA and NMDA synapses, the L2/3 PC axonal delay kernel is used and for GABAa and GABAb synapses the L2/3 BC axonal delay kernel is used. Each synaptic conductance is weighted by the surface area of the compartment, $A$, over the total surface area of compartments within the same bin, $A_{bin}$. The weight $f_{NMDA}$ is the fraction of NMDA synapses with respect to AMPA synapses (parametrizes the NMDA-AMPA balance) and the weight $f_{GABAa}$ is the fraction of GABAa synapses with respect to GABAb (parametrizes the GABAa-GABAb balance). The weight $f_{ex}$ is the fraction of all excitatory synapses with respect to inhibitory synapses (parametrizes the excitatory-inhibitory balance). The $f_{NMDA}$ and $f_{GABAa}$ free parameters ensure the total spike density from L2/3 PC is distributed between AMPA and NMDA synapses, while the total spike density from L2/3 BC is distributed between GABAa and GABAb synapses. The weights $f_{NMDA}$, $f_{GABAa}$, and $f_{ex}$ are free parameters whose values are chosen within ranges as described in the Sensitivity Analysis section. By defining synaptic inputs in this way, each compartment receives net input conductances (i.e., current) for each receptor type, defined directly by the axonal delay kernels and thus the spatio-temporal distribution of action potential arrivals at upstream axon terminals. Synapses were not discretely distributed across the dendritic tree, rather, synaptic inputs were defined by the spike density function of axon terminals whose positions spatially overlap with each dendritic compartment. Thus, synaptic inputs to L5 were defined directly by the L2/3 morphologies and their response to TMS induced electric fields.

Before simulation, the L5 PC firing dynamics are quenched by setting sodium channel conductance to zero on the axon and soma sections, such that only dendritic current due to synaptic input contributes to the current entering the soma. The

current entering the soma is the output measure of the synapto-dendritic delay simulation and the output of the whole coupling model (Fig 6I). It is derived from the quasi-1D cable equation (Eq (9)). The axial current along a cable $I_{axial}$ is calculated from the cross-sectional area $A$, the axial resistivity $R$, and the gradient of the membrane potential $V$ along the axis of the cable.

$$I_{axial} = -\frac{A}{R}\frac{dV}{dx} = -\sum_{i=1}^{N}\left(\frac{A_i}{R_i}\frac{V_{soma} - V_i}{L_i}\right)$$

(9)

If there are $N$ dendritic sections directly connected to the soma, the current is calculated as the sum of contributions from each section with index $i$ (Note that dendritic sections are distinct from segments in NEURON, where a section is a straight portion of morphology that may contain multiple segments). The membrane potential gradient for each adjacent dendritic section is calculated as the finite difference approximation between the potential at the soma, $V_{soma}$, and the potential at the center the of the dendritic section, $V_i$, divided by the path length distance from the center of the soma to the center of the adjacent section $L_i$. Each gradient is then scaled by cross-sectional area $A_i$ over the axial resistivity $R_i$. The finite difference sign is chosen such that current flowing into the soma is positive. We found, however, that even at steady-state when synaptic inputs were not present, a 0.109 nA background dendritic current flowed into the soma. This background current was subtracted from the results to ensure we only represent current due to synaptic inputs.

## Mean Field Calculation of Dendritic Current

Each synapto-dendritic delay simulation contains four fixed input parameters: $\theta$, $\Delta|\mathbf{E}|$, $|\mathbf{E}|$, and waveform type. Two parameters however are averaged across: Postsynaptic soma depth and postsynaptic morphology, such that the average current output of the dendritic delay simulations are computed as in Eq (10). To account for variability in the postsynaptic cell depth within the cortex, synapto-dendritic delay simulations are first averaged across $N_k = 25$ samples of soma depth, weighted by the normalized cell density for each depth value $w_k$ with index $k$ (Fig 7). Each weight is divided by the normalization constant, computed as the sum of cell densities for all depth values. Then, variability in dendritic geometry is accounted for by averaging across $N_{morph} = 30$ unique L5 PC morphologies. The result is an average current time series that is the output of the coupling model and represents the mean dendritic current a L5 PC population (or the average cell) receives due to lying downstream of L2/3 excitatory and inhibitory populations activated by TMS.

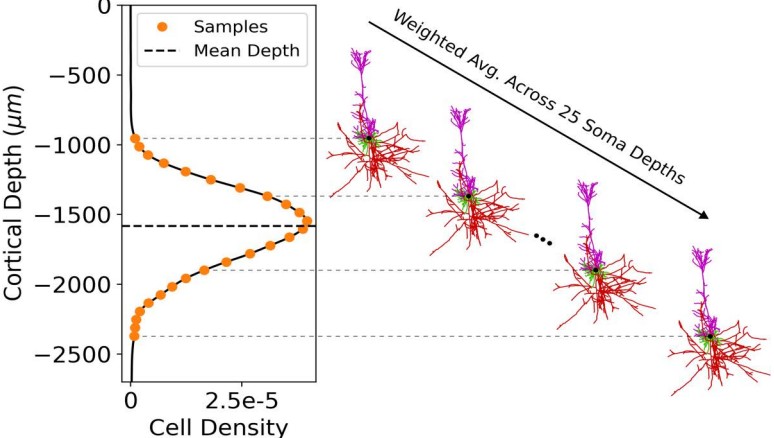

**Fig 7. Averaging algorithm of synapto-dendritic delay simulation across cortical depth, spanning the space occupied by L5 PCs. (left)** Cell density function of L5 PCs in M1 [28], overlaid with 25 equidistant samples in orange and the mean soma depth. For a given set of electric field and synapse distribution parameters, each synapto-dendritic delay simulation is computed and averaged for 25 different depths **(right)**, weighted by the cell density function for L5 PCs. The weight function is normalized such that the sum of weights across the 25 samples is 1.

$$\langle I \rangle = \frac{1}{N_{morph}} \sum_{j}^{N_{morph}} \left[ \frac{1}{\sum_{m}^{N_k} w_m} \sum_{k}^{N_k} w_k \, I_k(t) \right]$$

(10)

**Surrogate model of dendritic current.** For fast and efficient calculation of the dendritic current for different parameters of the applied electric field and type of synapse distributions, we used the generalized polynomial chaos method (gPC) to construct a surrogate model using the python implementation *pygpc* [34]. The gPC approximation allows a detailed time dependent sensitivity analysis in order to identify the most influencing parameters in the model. In order to provide a fairly general and flexible model, all parameters involved were defined within relatively broad bounds and modeled as uniformly distributed random variables. The TMS induced electric field parameters and their limits were the angle $\theta = [0, 180]°$, the relative gradient $\triangle|\mathbf{E}| = [-20, 20]$ %/mm, and the intensity $|\mathbf{E}| = [100, 400]$ V/m. The limits were motivated by observations from realistic head models from [23]. In addition, synapse parameters were also defined as free parameters within given limits, namely the fraction of NMDA synapses with respect to AMPA $f_{NMDA}=[0.25, 0.75]$ [35], the fraction of GABAa synapses with respect to GABAb $f_{GABAa}=[0.9, 1.0]$ [33], and the fraction of excitatory synapses with respect to inhibitory synapses $f_{ex}=[0.2, 0.8]$.

The gPC is based on modeling the input-output characteristic of black box systems using an orthogonal polynomial basis. For time dependent problems, a separate gPC is performed for each of the time points. The gPC surrogate of the dendritic current then reads:

$$I(t, \xi) = \sum_{i=1}^{N_c} c_i(t) \Psi_i(\xi),$$

(11)

where the vector $\xi$ contains the free parameters described previously. For each time point, $N_c$ gPC coefficients are determined independently. The maximum order of the polynomials $\Psi_i(\xi)$ is chosen to ensure the lowest possible approximation error while still being computational feasible. The maximum individual order of the polynomials was set to 20 if only one parameter is present in the basis function, e.g., in case of the polynomial for angle $\theta$, $\xi_\theta^{20}$, except for $\xi_{\triangle|\mathbf{E}|}$ describing the relative gradient of the electric field where the order could be reduced to six. The interaction order of the parameters $\xi$ was limited to three in the basis functions, allowing all possible parameter combinations of up to three between the six input parameters. If parameter combinations are present in the basis function, the maximum sum of all interacting polynomials was limited to 8 in order to reduce the number of underlying polynomials in the six dimensional space. The time axis $t$ was resampled to a time step of $\triangle t = 0.2$ ms in the range of 0 and 100 ms. This resulted in $N_c = 1642$ polynomial coefficients to be determined for each of the $N_t = 500$ time points after resampling. In order to determine the corresponding polynomial coefficients, $N_{sim} = 4000$ simulations were conducted with random parameter combinations within the specified parameter limits and the time curves of the dendritic currents were saved for each parameter combination. Summarizing all simulations, the gPC problem from Eq (11) can be formulated for all time points in matrix-form as:

$$[\mathbf{I}] = [\Psi] [\mathbf{C}],$$

(12)

where $[\mathbf{I}]$ is the current matrix of size $[4000 \times 500]$ containing the dendritic currents for all parameter combinations, $[\Psi]$ is the gPC matrix of size $[4000 \times 1642]$ containing all evaluated polynomials at the sampling points, and $[\mathbf{C}]$ is the coefficient matrix of size $[1642 \times 500]$ containing all unknown polynomial coefficients. The coefficient matrix is determined using the Moore-Penrose pseudoinverse of the gPC matrix, i.e., $[\mathbf{C}] = [\Psi]^+ [\mathbf{I}]$. The approximation error was evaluated by computing the average normalized root mean square deviation (NRMSD) over all time points using an independent test set of

$N_{test}$ = 1000 simulations with random parameter combinations. The average NRMSD between the gPC approximation and the original model was 1.9% and calculated as follows:

$$NRMSD = \frac{1}{N_t} \sum_{i=1}^{N_t} \frac{\sqrt{\frac{1}{N_{test}} \sum_{j}^{N_{test}} \left( i_j^{(ref)}(t_i) - i_j^{(gpc)}(t_i) \right)^2}}{\max(i^{(ref)}(t_i)) - \min(i^{(ref)}(t_i))},$$

(13)

where $i_j^{(ref)}(t_i)$ and $i_j^{(gpc)}(t_i)$ denote the original (reference) dendritic current and its gPC approximation for the $j$-th parameter combination in the test set at the $i$-th time point $t_i$, respectively. The gPC facilitates thorough analyses to assess how sensitive a system's output is to its input parameters and their uncertainties. This approach helps to identify the parameters that have the greatest impact on output variability. From the gPC coefficients, the Sobol indices and the global derivative-based sensitivity coefficients can be directly computed. Sobol indices decompose the total variance of the quantity of interest into contributions from individual parameters or their combinations [36,37]. On the other hand, the global derivative-based sensitivity coefficients are measures of the average change of the quantity of interest with respect to the parameters over the entire parameter space. Details about how the sensitivity measures are computed from the gPC coefficients are given in [38]. The 5000 dendritic delay simulations used to prepare the gPC were performed on 20 cluster nodes with 128 cores each (AMD EPYC Genoa 9554, 512 GB RAM) for 27 hours. In addition to the gPC, a direct lookup table of dendritic current for $f_{NMDA}$, $f_{GABAa}$, and $f_{ex}$ fixed at their average values was computed for 6 hours and is provided for reference.

## Results

As a case study motivated by DI-wave modeling, the coupling model is deployed to explore the TMS response functions of L2/3 excitatory and inhibitory cells as upstream populations, as well as the postsynaptic current in L5 cells as downstream populations receiving synapses from L2/3. First, the directional and dosage sensitivity of the axonal delay kernel, i.e., the direct spatial-temporal response to TMS, for L2/3 PCs is reported. Axonal delay is also shown for L2/3 BCs and L5 PCs in S1 Fig - S6 Fig. Then the directional and dosage sensitivity of the dendritic delay kernel, i.e., the somatic input current that modulates the neural state of L5 cells is reported. The Sensitivity Analysis section explores the sensitivity analysis of the dendritic delay kernel using gPC.

### Axonal delay Kernel: directional and dosage sensitivity

The axonal delay kernels for L2/3 PCs are reported below, where each is computed from histograms with bin sizes with $\Delta z$ = 100 $\mu$m and $\Delta t$ = 0.1 ms. They are interpolated using the piecewise cubic hermite interpolating polynomial method (pchip) [39] to d$z$ = 1 $\mu$m and d$t$ = 0.005 ms. The pchip interpolator preserves monotonicity in the interpolation data and prevents overshoot if the data is not smooth (Note that the dendritic delay kernel was computed with axonal delay kernels interpolated to 50 $\mu$m bins as in Fig 6).

The axonal delay kernel for the average L2/3 PC is shown in Fig 8 for five different electric field polar angles $\theta$ spanning 0° to 180° and four electric field intensities |**E**| between 175 V/m and 300 V/m that best capture near threshold activation up until saturation. At 175 V/m, near the cells' activation thresholds, spike density begins to appear weakly at $\theta$ = 0° and $\theta$ = 42°. At this intensity, a faint band of spike density about 0.3 ms$^{-1}$ $\mu$m$^{-1}$ strong is visible around 500 $\mu$m below the soma between 0.5 ms and 0.75 ms after stimulation begins. As intensity increases, cells are stimulated at all angles with increasing homogeneity. Additionally, more clusters of spike density appear including a 1-1.4 ms$^{-1}$ $\mu$m$^{-1}$ peak just below the soma around 0.6 ms, shifting closer to $t$ = 0 as intensity increases. Additional clusters of spike outputs all under 0.6 ms$^{-1}$ $\mu$m$^{-1}$ appear scattered above and far below the soma. These smaller peaks remain relatively constant despite minor shifts in shape and position and slowly increasing spike density as intensity increases. Higher spike density amplitudes at

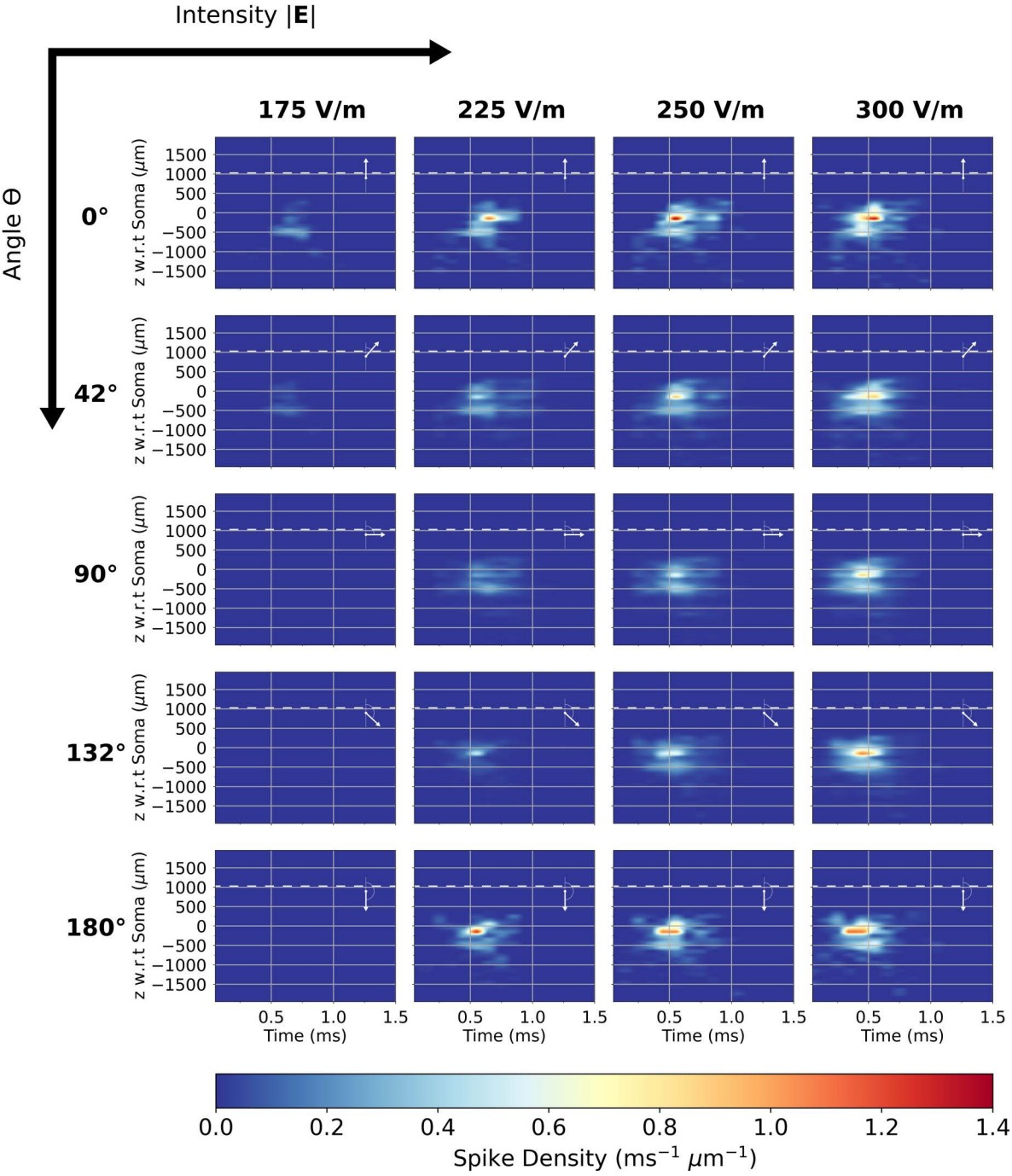

**Fig 8. Directional and intensity sensitivity of axonal delay kernel for the average L2/3 PC.** Kernels are shown in a grid with electric field intensity |**E**| increasing to the right and polar angle $\theta$ increasing to the bottom (electric field orientation w.r.t somatodendritic axis overlaid). Each kernel is plotted with respect to time and the depth with respect to cell soma. The color scale is spike density.

0° and 180° indicate that L2/3 cells send more spikes to downstream cells when electric fields are parallel or anti-parallel to the somatodendritic axis. The majority of synaptic outputs of L2/3 lay below the soma in two main bands, where spikes begin around 0.5 ms and vanish before 1 ms after the TMS pulse starts.

The axonal delay kernel for the distributed populations of L2/3 PCs is shown in Fig 9, and is equivalent to the kernels shown in Fig 8 correlated with the L2/3 cell density function from Fig 5D and multiplied by the L2/3 PC:L5 PC cell count ratio: 5.04. The correlation operation greatly smooths out the features of the average L2/3 PC output, as features that were formerly dependent on depth become smeared out along the vertical axis. The main cluster of spikes remains and exhibits overall the same directional and dosage sensitivity as in Fig 8 with a roughly 57% decrease in spike density amplitude. At 250 V/m and above, particularly for $\theta$ = 0° and 180°, an additional band of activity after the main peak appears around 0.8 ms.

Fig 10 also depicts the directional and dosage sensitivity of the L2/3 PC population, but with the cortical depth axis averaged out. By averaging the $z$-axis out of the kernels from Fig 9 the directional sensitivity of the spike outputs with respect to time is more visible. This collapse of the vertical axis could also prove useful for spiking neural network or mass modeling, in which point neurons and populations lack spatial extent. L2/3 PCs are initially recruited and outputting spikes near $\theta$ = 0°. At 225 V/m additional angles have been recruited with a gap around 135°, suggesting that recruitment occurs first at 0° and fills out higher angles. Between 175 V/m and 225 V/m, 180° is presumably recruited but before high angles like 135°. By 300 V/m, directional sensitivity is mostly vanished, reflecting that at this intensity roughly the same number of terminals are stimulated regardless of angle, as the electric field component parallel to most fibers is still sufficient to induce activation. As intensity increases further from 300 V/m, the peak in spike density curves shifts closer to $t = 0$ and becomes thinner and taller. This potentially reflects saturation in the total number of axon terminals that are activated (Fig 11A). As action potentials are elicited at more and more terminals, the expected amount of time it takes for terminals to receive an action potential decreases and becomes more synchronous. The backpropagation that carries action potentials to non-stimulated terminals at delayed intervals becomes less relevant as more and more axon terminals are directly activated. Additionally, the time it takes for an action potential to be elicited in any given axon terminal also shifts toward lower values as intensity increases (Fig 11B). The simultaneous effect of stimulating more terminals and stimulating terminals earlier results in spike density peaks that shift toward $t = 0$ and become more synchronous.

**Synapto-dendritic delay Kernel**

**Directional and dosage sensitivity.** The directional sensitivity of the fully combined coupling model (i.e., the surrogate gPC) is shown in Fig 12 with synapse free parameters set to: $f_{NMDA} = 0.5$, $f_{GABAa} = 0.9$ and $f_{ex} = 0.5$. Like the axonal delay kernels of L2/3 PCs and BCs, current is first visible near $\theta$ = 0° and begins to fill out higher angles. Current intensity also increases, and mostly all angles are recruited by 300 V/m. At 300 V/m a peak appears near $\theta$ = 0° which increases in amplitude. However, on average the amplitude of current does not continue to increase with increasing electric field intensity, suggesting that the role of inhibitory inputs causes a plateau if not decrease in dendritic current at some angles. The results suggest that currents received by downstream populations are notably directionally sensitive, inheriting this sensitivity from the direct TMS activation of upstream axons. The exact amplitude behavior of dendritic current is governed by the interplay of excitatory and inhibitory inputs from various source populations. Note that the secondary peak visible in the dendritic current timeseries was found to be a direct consequence of Shaw-related potassium channels on the soma and axons (by systematic silencing of channels), also appearing at the single simulation level.

**Sensitivity analysis**

The average dendritic current and variance of the gPC across all six free parameters ($\theta$, $\Delta|\mathbf{E}|$, $|\mathbf{E}|$, $f_{NMDA}$, $f_{GABAa}$, $f_{ex}$), with ranges reported in the Surrogate Model of Dendritic Current section, is shown in Fig 13A. The variance is considerable but reasonable given the large range of influential parameters like the electric field intensity. The dendritic current

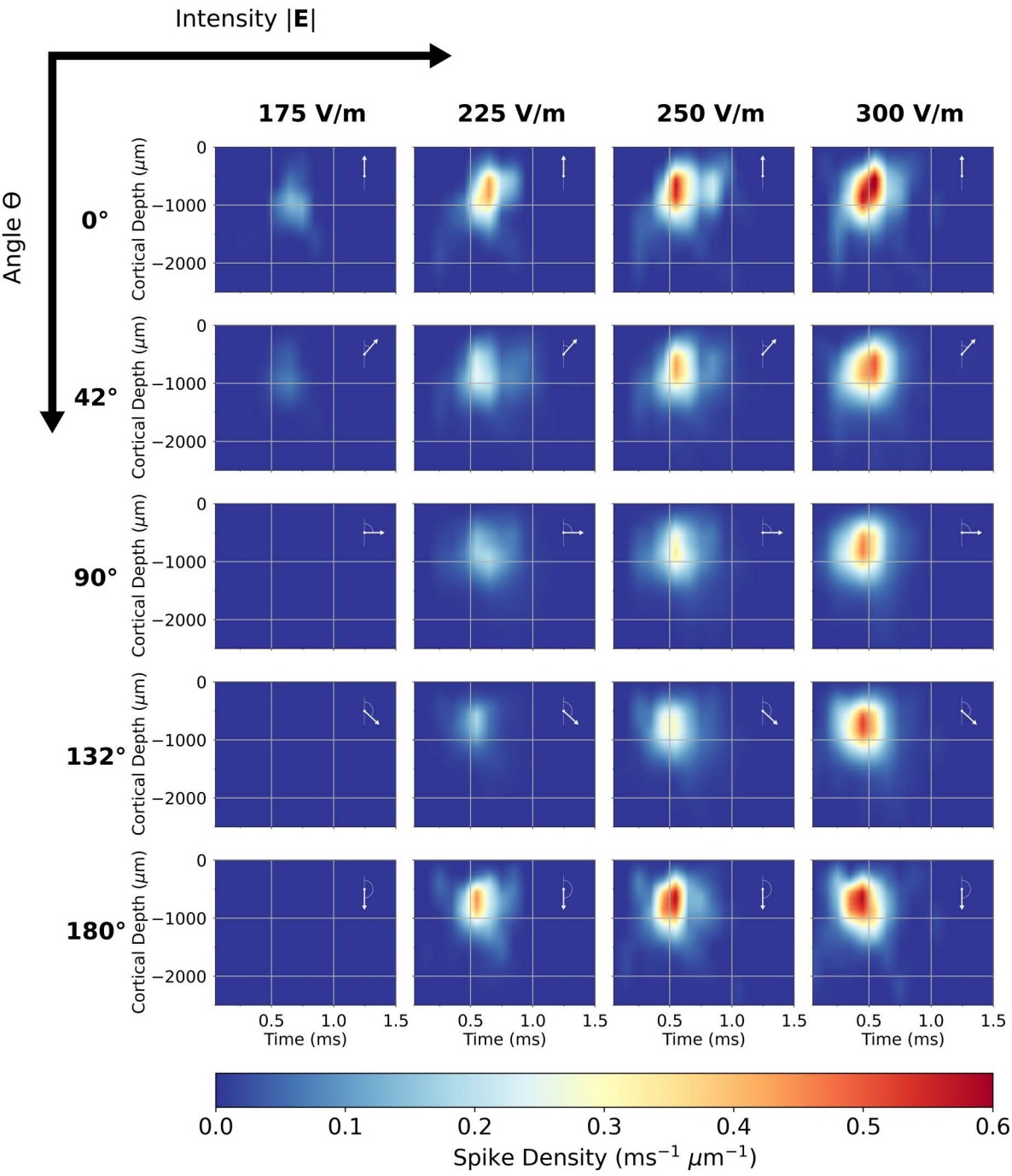

**Fig 9. Directional and intensity sensitivity of axonal delay kernel for the L2/3 PC population (Fig 8 after correlation with the L2/3 cell density function from Fig 5D and scaling by L2/3 PC:L5 PC cell count ratio 5.04).** Kernels are shown in a grid with electric field intensity |**E**| increasing to the right and polar angle $\theta$ increasing to the bottom (electric field orientation w.r.t somatodendritic axis overlaid). Each kernel is plotted with respect to time and the cortical depth with respect to CSF boundary. The color scale is spike density.

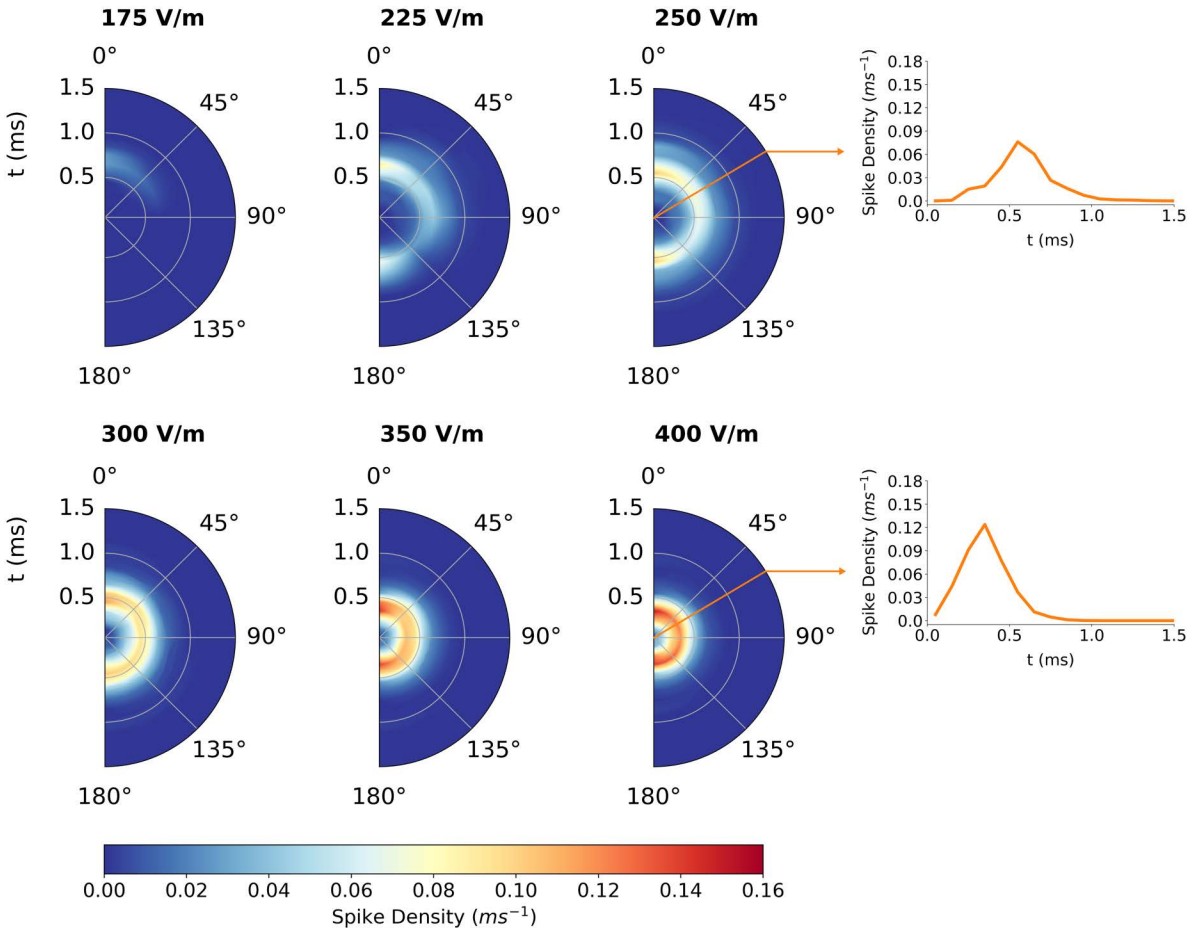

**Fig 10. Directional sensitivity of axonal delay kernel for L2/3 population with *z*-axis dependence averaged out.** Half polar plots for six electric field intensity |**E**| values included. 2D kernels as in Fig 9 are averaged across *z* to produce a time-dependent spike density for each angle. The radial axis of each half polar plot is time, the polar axis is the angle, $\theta$, the electric field vector makes with the somatodendritic axis, and the color scale is partial spike density.

timeseries indicates relative consistency in the shape and timescale of current input, consisting of a small initial peak followed by a larger broader peak whose height and decay is governed by the free parameters. The mean is also plotted for cases where the $f_{ex}$ parameter is fixed to 0.5 and 0.8 respectively, as a realistic excitatory/inhibitory balance in humans is around 0.8 to 0.9 [40]. The Sobol indices in Fig 13B and global derivative sensitivity measures in Fig 13C suggest the strongest influence of electric field intensity (|**E**|) and excitatory-inhibitory balance ($f_{ex}$) to the height and shape of the dendritic current. |**E**| plays a role mostly in the current within the first five milliseconds, with decaying influence up to thirty milliseconds. $f_{ex}$ alone, however, exerts little influence initially, but then plays a major role in the current amplitude from five milliseconds and on. The remaining parameters and parameter combinations have decreasing time-dependent significance on the timeseries. The combination of $\theta$ and |**E**| shows increased influence to the tail of the curve and also plays a larger role than $\theta$ alone, which reflects the coupling between directional sensitivity and electric field. The global derivatives confirm that $f_{ex}$, |**E**| and $\Delta$|**E**| serve to increase the current peak height. $f_{NMDA}$ acts to decrease the current peak, in which replacing some fast AMPA synapses with NMDA results in the activity under twenty milliseconds to be depressed.

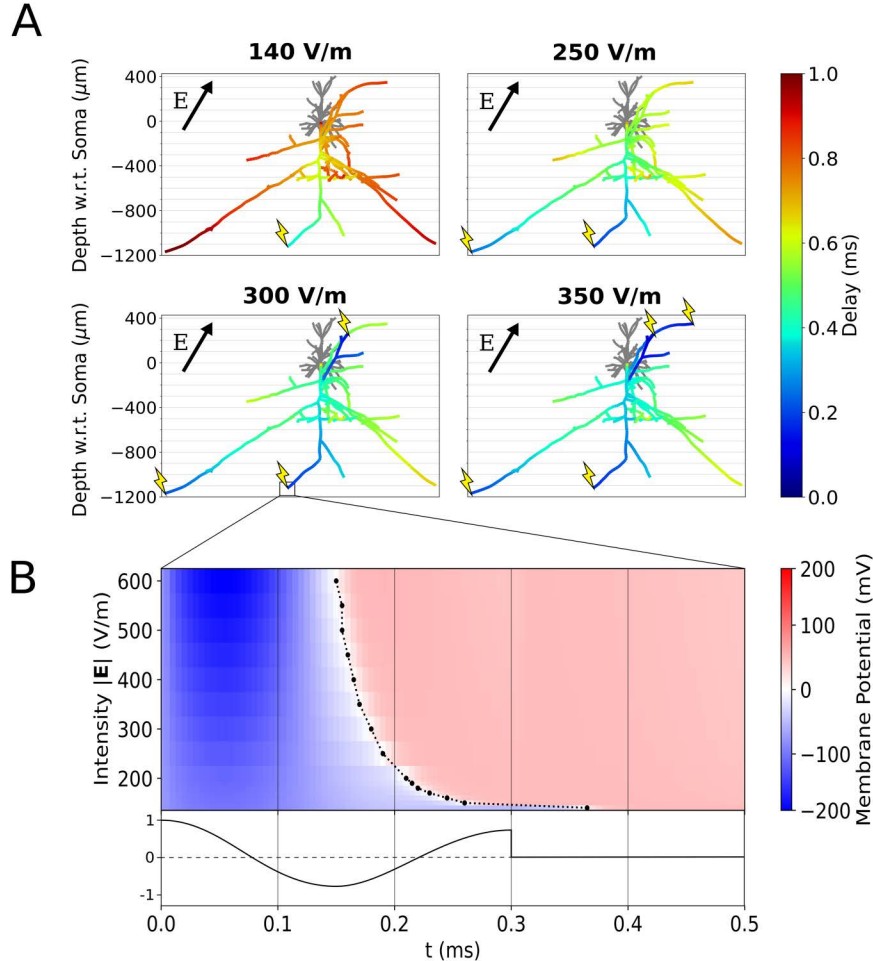

**Fig 11. Mechanisms of axon terminal recruitment saturation and spike output synchrony. (A)** Axonal delay simulations for a L2/3 PC at increasing electric field intensities (where $\theta = 30°$, $\Delta|\mathbf{E}| = 0\%/mm$, $\varphi = 0°$). The color axis quantifies the time delay after TMS at which an action potential reaches each axonal compartment. Lightning bolts indicate points of action potential generation. **(B)** Membrane potential at the boxed axon terminal compartment (the first terminal to be activated) as a function of time and electric field intensity. 0 mV is marked by a dashed line to show the shift of action potential generation time toward t = 0 as intensity increases. The biphasic TMS coil current waveform (normalized) is plotted beneath.

The time averaged Sobol indices in Fig 14 reaffirm the key role of excitatory-inhibitory balance, the electric field intensity, and the coupling between them to exert the largest influence on the dendritic current amplitude. Nonetheless, it is clear that the coupling between electric field angle and intensity as well as additional parameters and higher order combinations collectively play nontrivial roles.

## Discussion

Modeling of TMS offers important windows into the mechanisms which underlie neural processes, while aiming to eventually predict measurable readouts in response to stimulation. For the case of TMS of the primary motor cortex (M1), these readouts may include EEG, DI-Waves, and MEPs, which, in turn, provide measures against which to qualify and quantify model performance. This requires that the model realistically and comprehensively describes the relationship between these readouts and all relevant experimental parameters (coil geometry, position, and orientation, pulse shape and intensity, head geometry, etc.).

PLOS Computational Biology

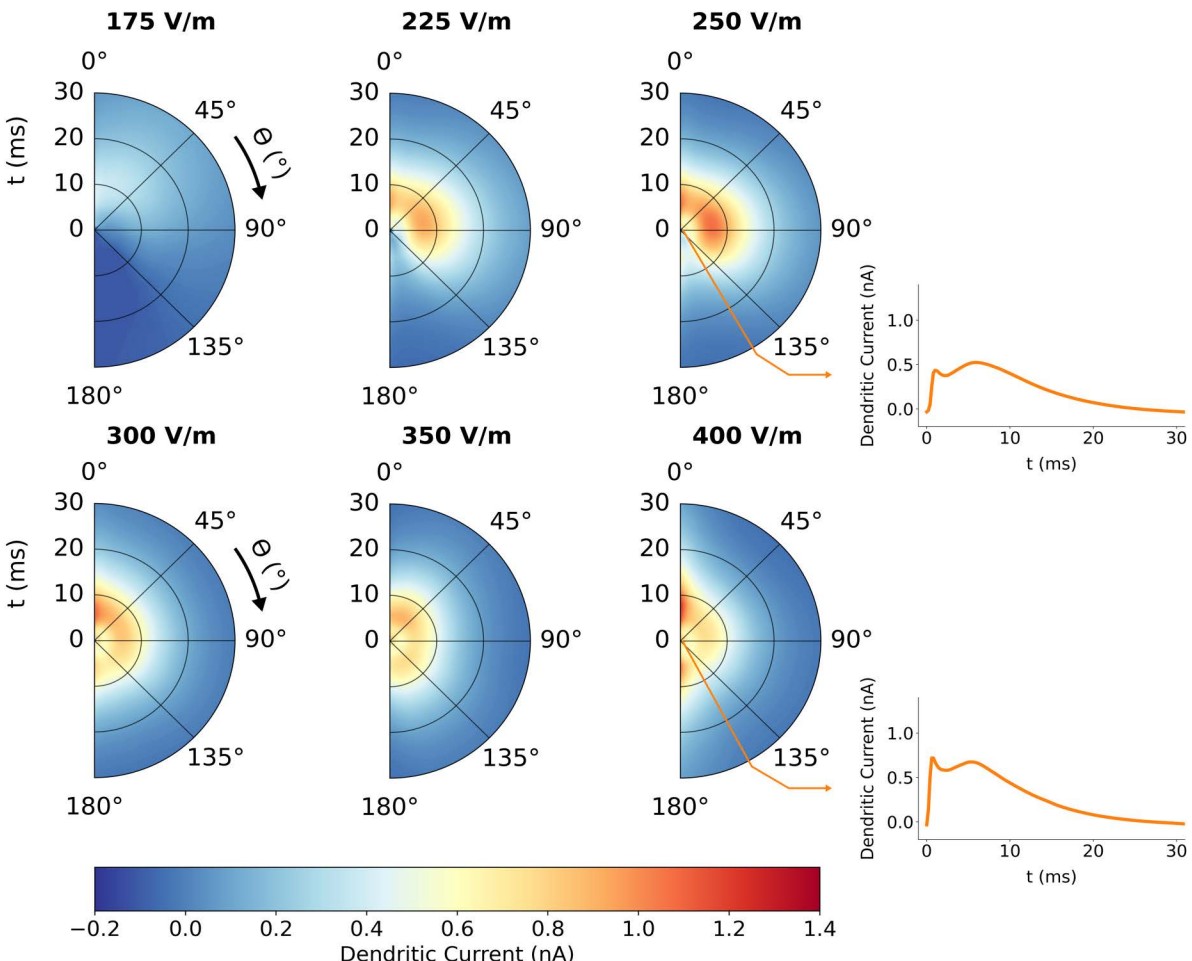

**Fig 12. Directional and intensity sensitivity of dendritic delay kernel.** The average dendritic current to L5 due to TMS activation of L2/3 PC and BC populations is plotted for six electric field intensities |**E**|. The radial axis of each half polar plot is time, the polar axis is the angle, $\theta$, the electric field vector makes with the somatodendritic axis, and the color scale is dendritic current in nanoamperes entering the L5 soma. Top right and bottom right are crosscuts showing average current when $\theta = 150°$ for |**E**| = 250 V/m and 400 V/m respectively.

Currently, an important missing link in this multi-step modeling chain is a realistic and efficient description of the coupling between TMS induced electric fields and the activation function that modifies neuron states. To fill this gap, this work develops a novel simulation pipeline in order to quantify the response of individual neurons to TMS, as well as couple this microscale interaction to macroscale variables like population firing rates and input currents.

Our pipeline comprises two principled steps, involving an upstream neuronal population whose axons are directly excited by the TMS induced electric field, sending synaptic output to a downstream population that generates the system's output, for example, by sending spike volleys down the axons of the spinal cord. The upstream neurons are described by the axonal delay kernel, which provides a spatial-temporal distribution of synaptic outputs that average cells or cell populations project to synaptic targets as a direct result of electromagnetic stimulation. This 2D kernel has been computed for various electric field orientations and intensities for different cell types, and will be made available for future works to reference. The behavior of the downstream or output neurons is described by the dendritic delay kernel, which provides the coupling between the synaptic output of a directly stimulated upstream population and the somatic input current of the

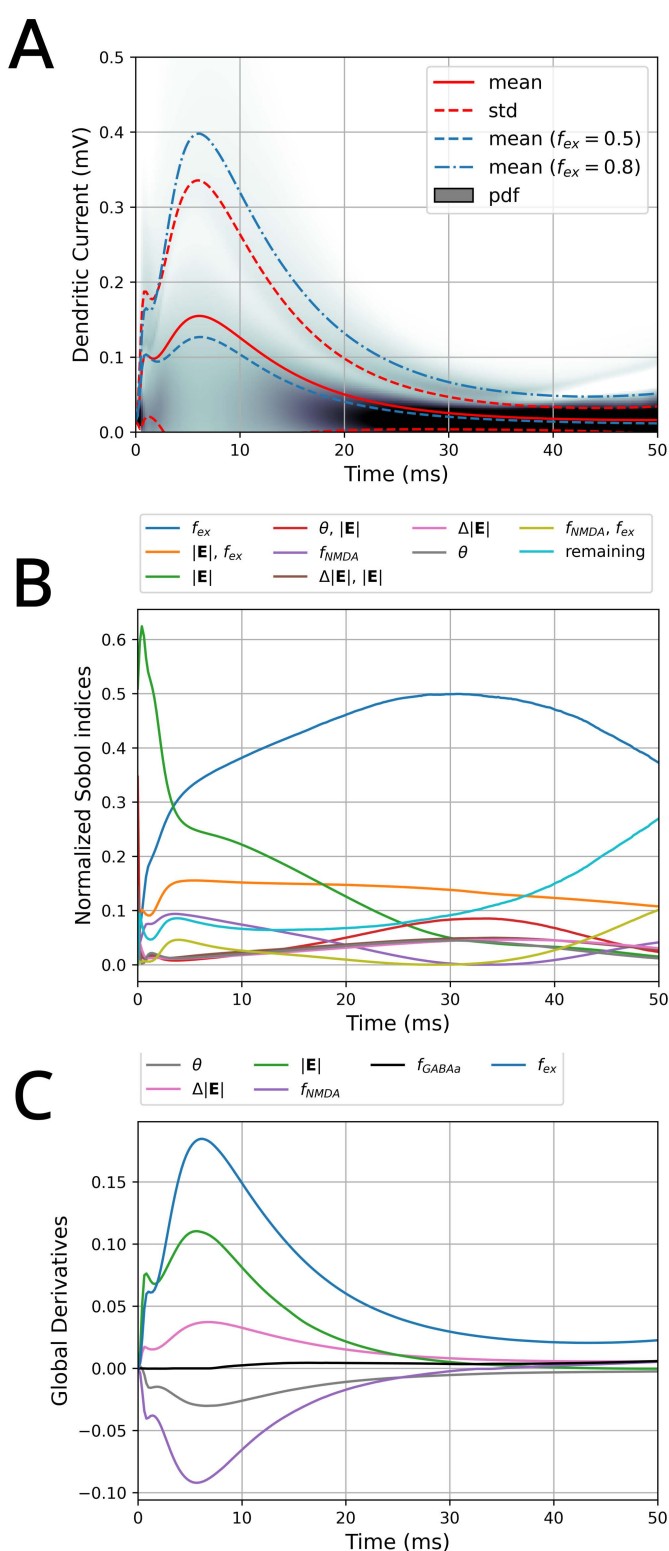

**Fig 13. Sensitivities of the dendritic current with respect to the input parameters over time. (A)** Average dendritic current $\pm$ its standard deviation; the grey shaded area shows the probability density. Depicted in blue are the average dendritic currents for cases when $f_{ex}$ is fixed to 0.5 and 0.8 respectively. **(B)** Highest normalized Sobol indices over time reflecting their relative contribution to the total variance. **(C)** Global average derivatives of the parameters over time.

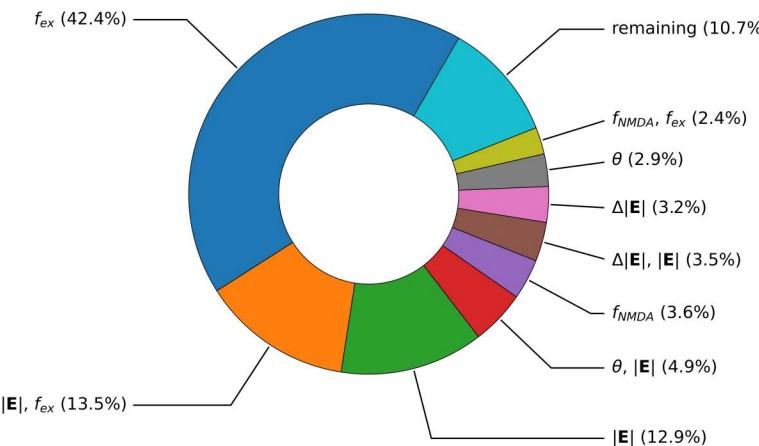

**Fig 14. Time averaged Sobol indices of the dendritic current.** (i.e., average of Fig 13B.).

downstream postsynaptic population. This current defines the state-modulating input from TMS and is applicable to drive cortical circuit models.

An example circuit for the prototypical architecture of M1 is then implemented and studied, with L2/3 excitatory and inhibitory neurons as upstream populations and L5 corticospinal neurons as the downstream population. The process of action potential backpropagation along axonal arbors, described by the axonal delay kernel, introduces unique spatial-temporal synaptic outputs for each cell type. These outputs are summed up in a non-linear fashion by the downstream dendritic tree, which is represented by the dendritic delay kernel. As a result, each combination of upstream and downstream cell types shows distinct directional sensitivity that is dependent on stimulation intensity. This sensitivity is particularly interesting in light of directionally sensitive experimental readouts of TMS. For example, recordings of I-waves during TMS suggest I-wave latencies and recruitment are sensitive to direction of TMS induced current [3,41–44]. If, for example, the cortical representation of the muscle target is on the sulcal wall, i.e., the relevant pyramidal cells are oriented largely parallel to the surface of the skull, rotation of the coil on the head surface could result in a rotation in electric field vector in the polar, $\theta$, direction. As L2/3 cells were shown to be directionally sensitive, coil rotation could induce different activation of L2/3 PC axon fibers. Hence, the interplay of directionally sensitive TMS activation of cortical axons with synaptic coupling in neural circuits in the cortex could help explain directionally sensitive behavior in TMS experiments.

As a consequence of the non-linearity of dendritic tree geometry, simulations of inputs from different upstream sources can not be computed separately and added together under the principle of superposition. Rather, all possible inputs must be simulated simultaneously to model interactions on the dendrite geometry. For example, [10] considers inputs to M1 from other nearby regions, like the premotor and somatosensory areas. Future study could also consider a TMS activation function specifically for long range fibers from other brain areas, and this would inform simulation of inputs to postsynaptic cells in M1.

Some key limitations of the coupling model extend from some structural assumptions and data availability. For example, as explored by [23], the electric field intensities that activate the compartment models studied here (around 170–350 V/m) are considerably larger than induced electric fields that are computed in the brain tissue for TMS intensities that activate M1 in experimental studies. However, neurons are not isolated in the cortex, but are embedded in tissue containing neurons, other types of cells, blood vessels, etc., all of which alter the microscale conductivity and the electric field a single neuron experiences locally. The influence of surrounding tissue on the activation of a single cortical neuron suggests that microscale conductivity gradients increase the electric field experienced by a neuron at its membrane surface

and thereby reduce the total applied macroscale electric field needed to activate the cell [45]. This result suggests that a correction factor or distribution could be introduced to rescale the macroscale parameterized electric field used in this coupling model to account for microscale effects.

Additional limitations extend from the use of reconstructed neuron morphologies from the somatosensory cortex. As morphologies were recovered from the somatosensory cortex but applied here for study of the motor cortex, it is possible that differences in axon and dendrite geometry skews the spatial-temporal activation function, the coupling between cells, and dendritic integration in a manner that is not accurate to the motor cortex. Nonetheless, these high quality cells from the Blue Brain Project [27] provide a good first approximation of cortical pyramidal and basket cells, but future study would benefit from reconstructed morphologies recovered from the motor cortex as well as comparison of the geometry and activation function of similar cells from different regions.

With respect to the synapse distribution algorithm, the synaptic coupling computed here defines synaptic inputs to the downstream population based on geometric overlap with axon terminals from the upstream population. In addition, we assumed that the axonal delay kernel is homogeneous within the x-y plane, i.e., that synaptic connections between layer 23 and layer 5 are random and uniformly distributed in the x-y plane. Some optical studies exist which estimate the spatial distribution of synaptic inputs, for example from L2/3 excitatory cells to L5 PCs in M1 [46]. However, no such data is available for the projection of L2/3 inhibitory cells to L5 PCs. As a result of this lack of data, and the desire to make the coupling model more independent of requiring detailed physiological experimental data, the synaptic coupling was rather formed intrinsically based on cell geometry and position.

Another key assumption of the synaptic inputs in this model is the electromagnetic stimulation of neurons at resting state, i.e., there is no background activity. Future work may also need to consider the effect of background inputs on the activation functions of cortical neurons.

The present framework provides a feedforward mapping from presynaptic population activity to synapto-dendritic input current and does not include activity-dependent feedback from the postsynaptic neuron. Action potentials generated during synaptic input backpropagate into the dendritic tree, modulate membrane voltage, and alter the state of voltage-gated ion channels and synaptic conductances, thereby influencing subsequent synaptic integration. These effects introduce nonlinear, state-dependent feedback that is not captured by the current synapto-dendritic delay kernel. Incorporating state-dependent feedback and recurrent dynamics would be necessary to capture spike-triggered dendritic nonlinearities and long-timescale network effects and represents an important direction for future extensions of the method.

Advantages of the coupling model, however, include modularity and the possibility for precomputation to reduce live simulation time. For example, the axonal delay kernel activation functions were calculated for three cell types and for many electric field parameters. Future studies of brain areas with comparable neuron morphologies would benefit from these activation functions precomputed to inform the TMS activation of point neurons, compartment models, or neural masses. Additionally, as the coupling model is generally defined along the synaptic projection between upstream and downstream populations, it is well suited for application to other brain regions with differing circuit architectures. The methods developed could also be applied to study other forms of non-invasive brain stimulation. Redefining the electric field input to the axonal delay kernel adjusts the model to reflect the unique manner in which other superthreshold magnetic stimulation scenarios generate action potentials in the cortex and couple to neural states.

In conclusion, the proposed framework provides a generic tool to couple TMS induced electric fields into dendritic currents entering the somata of cortical output neurons, thus determining their firing patterns. As many existing, otherwise sophisticated, models of the generation of short-latency cortical responses to TMS (i.e., DI waves) assume a very simplified relationship between action potential elicitation and electric field without any directional sensitivity (e.g., [10,14,15]), our framework offers a possibility to enhance these models without the need to carry out expensive single neuron simulations as in [19,20,23]. As the framework is made publicly available, it can be easily adapted to account for any circuitry of interest and to accommodate any newly available structural and physiological information. This coupling model allows for

the directional and dosing sensitivity of cortical neurons to inform the neural activation at the single cell or population level, such that future neural mass, point neuron network, and compartment modeling studies of TMS may define the effect of TMS based on biophysical microscale dynamics under electric fields.

## Supporting information

**S1 Video. Video of action potential backpropagation along L2/3 PC axonal arbor under biphasic TMS.** Time evolution of the membrane potential is depicted by color at each axon compartment, showing electric field induced action potential generation at one axon terminal, and the subsequent backpropagation to all remaining terminals. The membrane potential timeseries of the bottom right-most axon terminal (marked with a black dot) is depicted below.
(MP4)

**S1 Fig. Directional and intensity sensitivity of axonal delay kernel for the average basket cell.** Kernels are shown in a grid with electric field intensity $|\mathbf{E}|$ increasing to the right and polar angle $\theta$ increasing to the bottom (depiction of electric field orientation w.r.t somatodendritic axis overlaid top right of each kernel). Each kernel is plotted with respect to time and the depth with respect to cell soma. The color scale is spike density.
(TIF)

**S2 Fig. Directional and intensity sensitivity of axonal delay kernel for the inhibitory L2/3 BC population.** (S1 Fig after correlation with the L2/3 cell density function from Fig 5D and scaling by pre-:post-synaptic cell count ratio). Kernels are shown in a grid with electric field intensity $|\mathbf{E}|$ increasing to the right and polar angle $\theta$ increasing to the bottom (depiction of electric field orientation w.r.t somatodendritic axis overlaid top right of each kernel). Each kernel is plotted w.r.t. time and cortical depth from the CSF boundary. The color scale is spike density.
(TIF)

**S3 Fig. Directional sensitivity of axonal delay kernel for L2/3 inhibitory population with *z*-axis dependence averaged out.** Half polar plots shown for six electric field intensity $|\mathbf{E}|$ values. 2D kernels from S2 Fig are averaged across *z* to produce a time-dependent curve for each angle. The radial axis of each polar plot is time, the polar axis is the angle, $\theta$, the electric field vector makes with the somatodendritic axis, and the color scale is partial spike density. Top right and bottom right are crosscuts of the $|\mathbf{E}| = 250$ V/m and 400 V/m polar plots for $\theta = 60°$.
(TIF)

**S4 Fig. Directional and intensity sensitivity of axonal delay kernel for the average L5 PC.** Kernels are shown in a grid with electric field intensity $|\mathbf{E}|$ increasing to the right and polar angle $\theta$ increasing to the bottom (depiction of electric field orientation w.r.t somatodendritic axis overlaid top right of each kernel). Each kernel is plotted with respect to time and the depth with respect to cell soma. The color scale is spike density.
(TIF)

**S5 Fig. Directional and intensity sensitivity of axonal delay kernel for the L5 PC population (S4 Fig after correlation with the L5 cell density function from Fig 7).** Kernels are shown in a grid with electric field intensity $|\mathbf{E}|$ increasing to the right and polar angle $\theta$ increasing to the bottom. Each kernel is plotted with respect to time and the cortical depth with respect to CSF boundary. The color scale is spike density.
(TIF)

**S6 Fig. Directional sensitivity of axonal delay kernel for L5 population with z-axis dependence averaged out Half polar plots shown for six electric field intensity $|\mathbf{E}|$ values.** 2D kernels from S5 Fig are averaged across *z* to produce a time-dependent curve for each angle. The radial axis of each polar plot is time, the polar axis is the angle $\theta$ the electric

field vector makes with the somatodendritic axis, and the color scale is partial spike density. Top left is depicted a crosscut for $|\mathbf{E}|$ = 250 V/m, $\theta$ = 60°.
(TIF)

## Acknowledgments

The authors thank Torge Worbs for consultation and support in developing the model interface in NEURON.

## Author contributions

**Conceptualization:** Thomas R. Knösche, Konstantin Weise.

**Data curation:** Aaron Miller.

**Formal analysis:** Aaron Miller, Konstantin Weise.

**Funding acquisition:** Thomas R. Knösche, Konstantin Weise.

**Investigation:** Aaron Miller, Konstantin Weise.

**Methodology:** Aaron Miller, Thomas R. Knösche, Konstantin Weise.

**Project administration:** Thomas R. Knösche, Konstantin Weise.

**Resources:** Thomas R. Knösche, Konstantin Weise.

**Software:** Aaron Miller, Konstantin Weise.

**Supervision:** Thomas R. Knösche, Konstantin Weise.

**Visualization:** Aaron Miller, Konstantin Weise.

**Writing – original draft:** Aaron Miller, Thomas R. Knösche, Konstantin Weise.

**Writing – review & editing:** Aaron Miller, Thomas R. Knösche, Konstantin Weise.

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
