## [Decision Letter · Decision Letter 0]

3 Oct 2025

A Coupling Model of Transcranial Magnetic Stimulation Induced Electric Fields to Neural State Variables

PLOS Computational Biology

Dear Dr. Weise,

Thank you for submitting your manuscript to PLOS Computational Biology. After careful consideration, we feel that it has merit but does not fully meet PLOS Computational Biology's publication criteria as it currently stands. Therefore, we invite you to submit a revised version of the manuscript that addresses the points raised during the review process.

Please submit your revised manuscript within 60 days Dec 03 2025 11:59PM. If you will need more time than this to complete your revisions, please reply to this message or contact the journal office at ploscompbiol@plos.org. Please include the following items when submitting your revised manuscript:

We look forward to receiving your revised manuscript.

Kind regards,

Frédéric E. Theunissen

Academic Editor

PLOS Computational Biology

Daniele Marinazzo

Section Editor

PLOS Computational Biology

**Additional Editor Comments:**

Dear Konstantin,

Two expert reviewers have finished looking at you submission and had multiple comments that you will need to address. At a first glance, the lack of a capacitance current in your model seems like an important missing element but maybe there is a misunderstanding there.

Best,

Frederic Theunissen

**Journal Requirements:**

3) Some material included in your submission may be copyrighted. According to PLOSu2019s copyright policy, authors who use figures or other material (e.g., graphics, clipart, maps) from another author or copyright holder must demonstrate or obtain permission to publish this material under the Creative Commons Attribution 4.0 International (CC BY 4.0) License used by PLOS journals. Please closely review the details of PLOSu2019s copyright requirements here: PLOS Licenses and Copyright. If you need to request permissions from a copyright holder, you may use PLOS's Copyright Content Permission form.

Potential Copyright Issues:

- Figure 1. Please confirm whether you drew the images / clip-art within the figure panels by hand. If you did not draw the images, please provide (a) a link to the source of the images or icons and their license / terms of use; or (b) written permission from the copyright holder to publish the images or icons under our CC BY 4.0 license. Alternatively, you may replace the images with open source alternatives. See these open source resources you may use to replace images / clip-art:

4) Please ensure that the funders and grant numbers match between the Financial Disclosure field and the Funding Information tab in your submission form. Note that the funders must be provided in the same order in both places as well.

**Reviewers' comments:**

Reviewer's Responses to Questions

**Comments to the Authors:**

Reviewer #1: In this paper, the authors develop a pipeline for simulating the effects of transcranial magnetic stimulation (TMS) onto a postsynaptic population of cells. In particular, they consider the “indirect” effect of TMS on layer 5 (L5) pyramidal cells receiving inputs from L2/3 pyramidal and basket cells, which in turn are directly stimulated by TMS via activation of the axonal terminals that are aligned to the direction of the electric field generated by TMS stimulation. The paper is clear and the results that it presents are interesting: however, I think some changes should be made before the paper is suitable for publication in PLOS Computational Biology.

I have two main concerns about this paper, which are described in the following.

My first main observation has to do with the expectations that the authors build in the introduction, and that remain largely unmet at the end of the paper. In particular, in the introduction the authors describe the state of the art of TMS as it is used in a clinical setting, and present the full motor TMS cascade in Fig. 1: however, they then focus specifically on just one aspect of the full cascade, namely coupling the electrical field to the activation of neurons in L2/3 and L5 of the cortex. While there is nothing wrong with this (I realize that dealing with the whole pipeline as shown in Fig. 1 is well beyond the scope of pretty much any paper), and the authors clearly state it in the introduction, I think it would be beneficial if they toned down their claims and explained in more detail what the actual purpose of this paper is.

The second concern I have regards the practical applicability of the proposed pipeline: while the method looks intriguing, I don’t think the authors do a good job of explaining why it is superior (or alternative) to other techniques such as the ones described, for instance, in [1] and [2]. In the latter reference, in particular, the authors explicitly model D+ and D- responses and compare their results to experiments, something that would reside in panel D of Fig. 1 of the present manuscript. In order to be relevant, I think that new results should be an improvement over the current state of the art, rather than another way of doing something that might lead to potentially interesting results.

Other points that require some clarification are the following:

In Fig. 3B, does the time indicated in the caption (t = 0.525 ms) refer to the time of stimulation or is it the time of the frame?

1. Figure 4 would be more clear if the authors included some voltage traces corresponding to the different locations indicated with arrows and especially where the number of spikes is greater than one.

2. In line 234 they refer to Fig. 5A, B after the sentence “After averaging the axonal delay kernel histograms across each azimuthal angle, each cell morphology, and across cell subtypes for inhibitory basket cells”: however, Fig. 5 contains data related to pyramidal cells, and therefore I think that the sentence could be misleading. Please change it.

3. One of the conclusions they draw from Fig. 9 (lines 423-425) is that “L23 PC axonal arbor geometry is slightly anisotropic, such that more axon terminals are oriented parallel to the somatodendritic axis”: is this the case? In other words, can the authors test independently the degree of anisotropy of L2/3 PCs?

4. The difference between z axis and z’ axis is not clear.

Minor notes:

1. Line 236: “within in” should be just “within”.

2. Lines 253-254, the values of z coordinate should be accompanied by the units of measure, e.g., z = -2700 um.

3. In the caption of Fig. 6, there is a spurious “bf” before the label to panel (D).

4. Line 472, “soma” should be “some”.

5. Line 479, “underlay” should probably be “underlie”.

REFERENCES

[1] Aberra, A. S., Wang, B., Grill, W. M., & Peterchev, A. V. (2020). Simulation of transcranial magnetic stimulation in head model with morphologically-realistic cortical neurons. Brain stimulation, 13(1), 175-189.

[2] Yu, G. J., Ranieri, F., Di Lazzaro, V., Sommer, M. A., Peterchev, A. V., & Grill, W. M. (2024). Circuits and mechanisms for TMS-induced corticospinal waves: Connecting sensitivity analysis to the network graph. PLOS Computational Biology, 20(12), e1012640.

Reviewer #2: Reviewer summary

The authors present a methodology that acts as a surrogate to link action potentials generated by realistic electric field and neuron models to a transmembrane current that can serve as an input to other models, thereby bypassing the computational cost of simulating the realistic models. As a usage example, the authors applied this method to TMS activation of motor cortex and the generation of I-waves due to L2/3 activation, with applications toward coupling the activation to cortical network models.

The methodology is divided into two kernels. The axonal delay kernel empirically measures the spatio-temporal distribution of the spikes generated by coupling electric fields solved by the finite element method to morphologically-realistic, compartmental models of neurons with Hodgkin-Huxley style dynamics. The synapto-dendritic delay kernel convolves the spike densities of the axonal delay kernel with the conductance impulse responses of AMPAR, NMDAR, GABAaR, and GABAbR models, and takes into account axial propagation, to generate a singular current that represents the combined synaptic effect, due to TMS activation, that a postsynaptic soma would experience. By computing these currents for each neuron in a population of neurons, a more realistic coupling of TMS-induced electric fields to a network can be achieved.

In general, this methodology is an important step toward addressing a gap in the field, that is to realistically couple the suprathreshold effects of TMS to neuronal network populations. The methodology has an initial large computational cost to generate the axonal delay kernels, but the author’s plans to publish the kernels will alleviate the computational burden and serve to be an accessible tool.

However, there are three major concerns concerning the work. A line-numbered series of questions/comments follow these concerns.

Reviewer comments

1. The quasi-1D cable equation that determines the amount of current arriving at the soma does not include the capacitive current. To properly represent the passive propagation from dendrite to soma, the capacitive current needs to be included.

2. In general, the section describing the construction of the synapto-dendritic delay kernels needs some clarification. It is not clear from the text that the measurements were made directly in the L5 PC model. It is not clear that the kernel representations of the synaptic conductances were embedded throughout the morphology of the model, and simulations were run to measure the currents arriving at the soma. My initial interpretation was that the kernels were distributed according to the coordinates of the morphology but were not in actual mechanisms within the membrane, and the quasi-1D cable equation would only account for the passive properties of the dendrites. Rather, because the currents are propagated through the full model, the output current accounts for the passive and active properties of the dendrites and the morphological structure. This was only apparent after going through the github code. This section needs to be rewritten to make this more clear. Some of the line-numbered comments fall under this theme.

3. Because the method is an approximation of coupling of the finite element derived electric field to a neuronal population, there should be some characterization of how well the full model is approximated. How well does the axonal delay kernel capture the spatio-temporal distribution of spikes generated by a full finite-element / population model? When a L5 PC neuron is driven by the synapto-dendritic delay kernel output, how does its somatic membrane voltage differ from the case when its actual synapses are driven by the axonal delay kernel’s spike densities (current injection from synapto-dendritic delay kernel vs synaptic activation due to suprathreshold axon terminals)?

Fig 4: Please add axis labels on left subplot.

Lines 215, 302-303: The axonal delay kernel accounts for the z-position of the arrival/initiation of the action potential relative to the soma. Based on how the axonal delay kernel is linked to the synapto-dendritic delay kernel, it appears that the positions of the x- and y-positions of axon terminals relative to the x- and y-positions of the dendrites are not taken into account, e.g. regardless of the x- and y-distances of the dendritic compartment relative to the axon terminal, the latency of synaptic activation for dendritic compartments in the same z-location were identical. How much would including x- and y-distances impact the temporal properties of the axonal delay kernel?

Line 277: Please provide a brief description of the pchip interpolation method.

Line 287: Which L5 PC morphology was used to extract the synapto-dendritic delay kernel?

Fig 6G and H: How are the synaptic conductances converted into a current? Where does the voltage in Eq 2 come from?

Line 295: How were the parameters in Table 2 determined? Were they chosen to fit EPSC properties such as peak current, time-to-peak, half-height width, etc?

Line 302: “Each synaptic conductance convolves the double exponential kernel…” This should read “Each synaptic conductance is computed by convolving the double exponential kernel with the partial integral of…”

Line 308: What were the total numbers of synapses that the L5 PC neurons had (AMPAR + NMDAR + GABAaR + GABAbR)? What discretization value was used for their morphologies (compartment size)? How does the choice of discretization value affect the final current output of the synapto-dendritic delay kernel? How were synapses distributed through the dendrites, e.g. evenly across all dendritic compartments or heterogeneously according to spine densities that can be dependent on the proximodistal location relative to soma?

Line 317: What other voltage-gated ion channels were present in the models? Setting sodium conductances to zero would only eliminate sodium currents and only if the voltages of all compartments were at the equilibrium potential of the sodium channel. If there are other voltage-gated ion channels, then they will also contribute current to the soma if their conductances are not set to zero. The models should be simulated until the voltages of all compartments reach their steady-state. If the currents are non-zero at steady-state, then the non-zero value should be offset from the final kernel output.

Lines 320–323; Eq 9: The synapto-dendritic delay kernel takes into account the effect of distance between the dendritic compartment and the somatic compartment using the quasi-1D cable equation but only takes into account the resistive current; the capacitive current is absent. To capture the total passive effect of the somato-dendritic integration, the capacitive current should also be included.

Line 324: “If there are N dendritic sections connected to the soma compartment, the current is calculated as the sum of contributions from each section with index i.” Is the current calculated only from dendritic compartments directly connected to the soma, all dendritic compartments, or all dendritic compartments that have synapses? In the parlance of NEURON, are segments or sections being used to determine adjacency to the soma?

Lines 327-328: Is distance between dendritic compartment center and soma computed using the path distance along the morphology or the Euclidean distance between the coordinates?

Line 344: Please provide a brief description and/or general formula for the generalized polynomial chaos method.

Lines 358–361: What does it mean that the maximum order of individual polynomials was set to 20? Can certain time points have a different polynomial order than others? What does it mean that maximum sum of all interacting polynomials was limited to 8? Please clarify how the orders of the polynomials were determined.

Lines 455–456: Can the surrogate gPC model for the synapto-dendritic kernel be separated to compare the sensitivity analyses for L2/3 PC based (excitatory) activation vs L2/3 BC based (inhibitory) activation? Do these cell types exhibit different variances in dendritic current and directional sensitivities? Do different basket cell classes exhibit different properties as well?

Fig 12: What contributes to the two peaks in the dendritic current? There is a sharper peak around 1 ms and a second broader peak around 6 ms.

The amplitude of the dendritic current remains large up to 10 ms. Given that typical I-wave intervals can be 1-1.5 ms, this sustained current could potentially trigger 6-10 I-waves could occur during this time window. However, 6-10 I-waves are rarely observed in patients. What contributes to this discrepancy?

Line 523: Another limitation to include is that the method is strictly feedforward. Any action potentials that are generated during the injection of current produced by the synapto-dendritic delay kernel by the postsynaptic neuron will backpropagate through the dendrites and should affect the state of the synapses by modifying the membrane voltage, changing the state of the voltage-gated ion channels, etc. However, the present form of the methodology does not take into account the state of the neuron as the current drives the model (feedback).

**Have the authors made all data and (if applicable) computational code underlying the findings in their manuscript fully available?**

The PLOS Data policy requires authors to make all data and code underlying the findings described in their manuscript fully available without restriction, with rare exception (please refer to the Data Availability Statement in the manuscript PDF file). The data and code should be provided as part of the manuscript or its supporting information, or deposited to a public repository. For example, in addition to summary statistics, the data points behind means, medians and variance measures should be available. If there are restrictions on publicly sharing data or code —e.g. participant privacy or use of data from a third party—those must be specified.requires authors to make all data and code underlying the findings described in their manuscript fully available without restriction, with rare exception (please refer to the Data Availability Statement in the manuscript PDF file). The data and code should be provided as part of the manuscript or its supporting information, or deposited to a public repository. For example, in addition to summary statistics, the data points behind means, medians and variance measures should be available. If there are restrictions on publicly sharing data or code —e.g. participant privacy or use of data from a third party—those must be specified.

Reviewer #1: Yes

Reviewer #2: Yes

PLOS authors have the option to publish the peer review history of their article (what does this mean? ). If published, this will include your full peer review and any attached files.). If published, this will include your full peer review and any attached files.

.

Reviewer #1: **Yes:** Daniele LinaroDaniele Linaro

Reviewer #2: **Yes:** Gene J. YuGene J. Yu

**Figure resubmission:**
---

## [Decision Letter · Decision Letter 1]

6 Mar 2026

Dear Dr Weise,

We are pleased to inform you that your manuscript 'A coupling model of transcranial magnetic stimulation induced electric fields to neural state variables' has been provisionally accepted for publication in PLOS Computational Biology.

Best regards,

Frédéric E. Theunissen

Academic Editor

PLOS Computational Biology

Daniele Marinazzo

Section Editor

PLOS Computational Biology

Dear Authors,

Please make sure to address the minor points of Reviewer 2 in your final submission. Congratulations on your contribution.

Frederic Theunissen.

Reviewer's Responses to Questions

**Comments to the Authors:**

Reviewer #1: The authors have satisfactorily addressed all my comments and therefore I recommend the manuscript for publication in PLOS Computational Biology.

Reviewer #2: The authors have clarified my understanding of the work through their responses to my prior comments. Their additions to the manuscript address my concerns. I’d like to emphasize again the utility of this work as a tool to link the output of highly-detailed and complex simulations involving electric fields and single neurons to activation of neuronal networks.

A general theme of my previous comments pertained to the clarity of the description of how the synapto-dendritic kernels were computed. My misconceptions, e.g., capacitive currents, voltage used in Ohm’s law, etc., were based on a misunderstanding that the output of the synapto-dendritic kernels was being computed outside of NEURON. Rather, it was only the inputs to the simulations for measuring the synapto-dendritic kernels, i.e., the calculation of synaptic currents based on the output of the axonal delay kernels, that was computed outside NEURON. With the authors’ responses and clarifications in the text, it is now clear that the synapto-dendritic kernels were ultimately constructed using NEURON simulations, and so a majority of my concerns have been addressed.

Below are some minor comments that should be considered before the final manuscript is submitted.

1. Another theme of my previous comments was about the distribution of synapses and morphological discretization of the L5 PCs. The author’s responses have clarified that the total current that drives each compartment is dependent entirely on the distribution of axon terminals due to rotating and shifting the morphologies of the presynaptic populations. Independent of the discretization of the postsynaptic morphology, multiple axon terminals can impinge onto the same z-extent to drive putative synapses. The authors noted in their response that “… we discovered that the model was highly sensitive to the choice of discretization…”. However, a discretization must have been selected to execute the simulations for measuring the synapto-dendritic kernels. Please report simply the discretization used in the models.

2. The fex for cortex is relatively consistent at approximately 0.8-0.9 (see Table 2 in ref below). The authors should consider adding a line to Fig. 13A that indicates the response at this value as a putative response for a realistic excitatory/inhibitory ratio. This could be contrasted with the value at fex=0.5 whose directional/intensity sensitivity was shown previously.

DeFelipe, J., Alonso-Nanclares, L., & Arellano, J. I. (2002). Microstructure of the neocortex: Comparative aspects. Journal of Neurocytology, 31(3–5), 299–316.

3. Line 191-192: What time step size (dt) was used to reach steady state during the 10^11 ms simulations?

4. The Fig. 6 caption states that a 0.025 ms step size was used for the synapto-dendritic delay simulations. This should be stated in the main text instead.

5. Fig 5C: Should L2/3 PC soma be at 0 w.r.t to the y-axis tick marks? The dashed horizontal line is around 1000.

6. Fig 5D caption description: Reference to 5B should be 5C

7. L23 should be L2/3 throughout the manuscript and figures.

**Have the authors made all data and (if applicable) computational code underlying the findings in their manuscript fully available?**

The PLOS Data policy requires authors to make all data and code underlying the findings described in their manuscript fully available without restriction, with rare exception (please refer to the Data Availability Statement in the manuscript PDF file). The data and code should be provided as part of the manuscript or its supporting information, or deposited to a public repository. For example, in addition to summary statistics, the data points behind means, medians and variance measures should be available. If there are restrictions on publicly sharing data or code —e.g. participant privacy or use of data from a third party—those must be specified.requires authors to make all data and code underlying the findings described in their manuscript fully available without restriction, with rare exception (please refer to the Data Availability Statement in the manuscript PDF file). The data and code should be provided as part of the manuscript or its supporting information, or deposited to a public repository. For example, in addition to summary statistics, the data points behind means, medians and variance measures should be available. If there are restrictions on publicly sharing data or code —e.g. participant privacy or use of data from a third party—those must be specified.

Reviewer #1: Yes

Reviewer #2: Yes

PLOS authors have the option to publish the peer review history of their article (what does this mean? ). If published, this will include your full peer review and any attached files.). If published, this will include your full peer review and any attached files.

.

Reviewer #1: **Yes:** Daniele LinaroDaniele Linaro

Reviewer #2: **Yes:** Gene J. YuGene J. Yu

---

## [Editor Report · Acceptance letter]

PCOMPBIOL-D-25-01613R1

A coupling model of transcranial magnetic stimulation induced electric fields to neural state variables

Dear Dr Weise,

I am pleased to inform you that your manuscript has been formally accepted for publication in PLOS Computational Biology. Your manuscript is now with our production department and you will be notified of the publication date in due course.

With kind regards,

Zsofia Freund
